# Anapole-state-enhanced 2D chiral photodetector operating in the near-infrared second window

Qi-hang Zhang [1,2,3,4,7], Zi-hao Dong[1,2,3,4,7], Kai Liu[1,5], Shao-jie Fu[1,5], Xu-hao Hong[1,5], Yu-lin Cao[6], Chao Zhang[1,2,3,4], Jun Du[1,5], Yan-qing Lu [1,2,3,4], Yong-yuan Zhu[1,5], Yan-feng Chen [1,2,3,4] & Xue-jin Zhang [1,2,3,4] ✉

Two-dimensional (2D) materials hold promise for miniaturized photodetectors. With ample exciton resonances, the photodetection range of transition metal dichalcogenides (TMDCs) can be further extended to long wavelengths on a large scale by two-photon absorption (TPA), breaking the limit of their bandgaps. However, the conversion efficiency of TPA usually remains low despite resonant nonlinear optical effects. Here, we present a plasmonic metasurface-enhanced 2D TMDC photodetector by means of high-order multipoles with anapole states, as well as quasi-bound states in the continuum, operating efficiently in the near-infrared second (NIR-II) window at room temperature. The optical response of the $MoS_2/WSe_2$ heterostructure is simultaneously enhanced by the interlayer exciton resonances and by the hot carrier injection from the plasmonic metasurface. By optimizing the metasurface design, the responsivity can reach 1.35 A/W at 1550 nm, which is ~$5 \times 10^4$ times larger than that of a $MoS_2/WSe_2$ heterostructure on $SiO_2/Si$ substrate. Furthermore, the broken mirror symmetry of the structure enables a chiral photoelectric response with discrimination ratios up to 7.2. Our study offers a promising platform for applications in NIR-II bio-imaging, telecommunication, and on-chip spectroscopic sensing.

The relentless advancement of photodetection technology is driven by the demand for devices that are not only highly sensitive and broadband but also multifunctional and compact. While conventional semiconductors like Si and InGaAs have dominated the landscape, their inherent limitations in spectral region, integration density, and mechanical rigidity have spurred the exploration of alternative materials. The advent of two-dimensional (2D) materials has introduced a paradigm shift, offering atomic-scale thickness, exceptional mechanical flexibility, and strong light-matter interactions[1–4], thereby presenting a compelling platform for next-generation miniaturized and wearable photonic and optoelectronic devices[5–7].

Among the 2D family, transition metal dichalcogenides (TMDCs) have garnered significant interest due to their direct bandgaps in the monolayer form, large exciton binding energies, and valley-dependent optical properties[8,9]. In particular, being endowed with diversified excitons, TMDCs possess strong light-matter interaction and high quantum efficiency, leading to high-performance photoelectric devices. Despite these advantages, the

[1]State Key Laboratory of Solid State Microstructures, and Collaborative Innovation Center of Advanced Microstructures, Nanjing University, Nanjing, China. [2]Key Laboratory of Intelligent Optical Sensing and Manipulation, Nanjing University, Nanjing, China. [3]Jiangsu Key Laboratory of Artificial Functional Materials, Nanjing University, Nanjing, China. [4]College of Engineering and Applied Sciences, Nanjing University, Nanjing, China. [5]School of Physics, Nanjing University, Nanjing, China. [6]Physics Laboratory, Industrial Training Center, Shenzhen Polytechnic University, Shenzhen, China. [7]These authors contributed equally: Qi-hang Zhang, Zi-hao Dong. ✉e-mail: xuejinzh@nju.edu.cn

photoresponse of most TMDC-based devices is fundamentally constrained to the visible range by their electronic bandgaps[10,11], which severely limits their utility in the near-infrared second (NIR-II) window (1000–1700 nm), which is critical for biomedical imaging[12,13], fiber-optic communications[14], and surface-enhanced Raman scattering sensing[15,16]. The engineering of van der Waals heterostructures (e.g., MoS$_2$/WSe$_2$)[17–19] can form interlayer excitons, slightly expanding spectral response range. Fortunately, the exploitation of nonlinear optical processes such as two-photon absorption (TPA)[20–24] can detect sub-bandgap photons by a large margin. However, the inherently weak nature of TPA results in low responsivity, posing a formidable challenge for practical applications.

The integration of plasmonic nanostructures with semiconductors represents a powerful and generic approach to overcome the limitations of low optical absorption and weak nonlinearities in thin materials[25–27]. By confining light into sub-diffraction-limited volumes and generating intense localized electromagnetic fields, plasmonic modes such as surface plasmon polaritons (SPPs), localized surface plasmon resonances and gap plasmon polaritons (GPPs)[26,27], can dramatically enhance light-matter interactions. Furthermore, the decay of these plasmons generates highly energetic 'hot carriers'[28,29], which can be injected into adjacent semiconductors, providing a photocurrent gain mechanism that is independent of the semiconductor's bandgap[30]. Recent advances in metasurfaces, composed of precisely engineered meta-atoms, have unlocked unprecedented control over photoelectric behavior at the nanoscale[31–33]. Nonradiative schemes such as anapole states[34–37] and bound states in the continuum (BICs)[38–41] offer unique pathways to confine light and manipulate its scattering properties[33,34], giving rise to extreme field enhancement[37,39] with suppressed radiative losses. Unlike conventional optical elements, metasurfaces can manage the phase[42], amplitude[39,40], and polarization[43] of light at a deep subwavelength scale through carefully engineered geometries. This capability allows for the creation of devices with built-in functionalities such as chiral sensing[44], moving beyond simple photodetection towards advanced on-chip polarimetric and spectroscopic systems[45].

In this work, we converge these advanced concepts by integrating a MoS$_2$/WSe$_2$ van der Waals heterostructure with a meticulously designed plasmonic metasurface fabricated on low-loss single-crystalline Ag. We strategically employ anapole states, as well as quasi-BICs, to activate high-order multipoles, resulting in remarkable electromagnetic field enhancement across the NIR-II spectrum. A local electric field enhancement ($E_{enhanced}/E_{input}$) of >1000 has been achieved following this mechanism[46], which is an unexpected value in an open architecture. This substantial enhancement subsequently led to exceptional nonlinear optical performance[47]. It not only amplifies the TPA process within the TMDC heterostructure but also enables efficient injection of plasmon-generated hot holes, thereby bypassing the traditional bandgap limitation[29,30]. The augmented generation of hot holes stems from directly field enhancement at the excitation wavelength of NIR-II region and the recycling process of exciton energy, which dissipated into electron-hole excitations and interface plasmonic modes[29]. These give birth to a monumental improvement of ~5 × 10$^4$-fold over the TMDC heterostructure alone. In consequence, a responsivity of 1.35 A/W at 1550 nm is obtained, comparable to commercial one. In addition, by controllably breaking the mirror symmetry of the metasurface, strong polarization sensitivity is encoded into the photoresponse, achieving considerable chiral discrimination. This study exemplifies a comprehensive design strategy where nanophotonic engineering is employed to achieve unparalleled responsivity and spectral range, together with advanced functionalities like polarization resolution, towards intelligent, multi-functional photodetectors in a highly integrated form factor.

## Results and discussion

### Basic structure and properties of the nanostructured device

Figure 1a illustrates the layout of the photodetector (see Supplementary Fig. 1 for fabrication details). The plasmonic metasurface consists of an array of cruciform structures, which is etched on the single-crystalline Ag surface by focused ion beam (FIB), as shown in the Fig. 1b, together with the scanning electron microscope (SEM) image of the metasurface. The resonance wavelength can be adjusted by geometric parameters. For the structure, a GPP mode can be excited under horizontally polarized illumination. In order to achieve broadband enhancement, the array period is set as $P = 1000$ nm (Supplementary Fig. 2). Considering the performance of the metasurface at the wavelength of 1550 nm, the parameters are optimized as $L_V = 800$ nm, $w = 100$ nm, $d = 300$ nm, and the horizontal groove is infinite, as shown in Supplementary Fig. 2,3. The asymmetry of structure, $\Delta y$, is another degree of freedom to control field confinement. At $\Delta y = 0$, the structure supports a BIC mode (Supplementary Fig. 5), which is decoupled from far-field radiation and thus cannot be directly excited[38,39]. With zero radiative loss, it cannot be utilized directly. When $\Delta y \neq 0$, the symmetry is broken, introducing radiative losses that transform the ideal BIC into a leaky quasi-BIC, which can exchange energy with outside, manifested as certain linewidth in reflection spectra[43]. With the increase of $\Delta y$, two modes gradually split up in the frequency domain, as shown in Supplementary Fig. 4a, b. The $Q$ factor of the quasi-BIC decreases with $\Delta y$ while that of the GPP mode shows opposite behavior, which provides pathways to regulate the radiative loss and maximize the field enhancement[39]. Supplementary Fig. 4c shows that the maximum performance of the quasi-BIC occurs at $\Delta y = 100$ nm. The simulated reflection spectrum of the structure with $\Delta y = 100$ nm is shown in Fig. 1c, with the mode wavelength of the quasi-BIC is fixed at 1550 nm, which is consistent with the measured one. It can be found that the electric field intensity is concentrated on the corner of the structure from the inset of Fig. 1c, which indicates a small mode volume and strong field confinement, yielding a local field enhancement factor (EF) exceeding 800. To gain insight into the physical mechanism, multipole decomposition[48,49] of the quasi-BIC is carried out, as shown in Fig. 1d. Abnormally, a higher-order multipole, electric quadrupole (EQ) occupies a large proportion, which brings smaller mode volumes, more beneficial to the field enhancement[46]. Tracing its origin, the electric dipole (ED) destructively interferes with the toroidal dipole (TD), creating anapole states (see Supplementary Fig. 6 for details), which do not radiate to the far field and conserve energy in the near field[37]. Then, the higher-order multipoles dominate the optical mode, largely boosting the electromagnetic field enhancement[46]. This suggests that the quasi-BICs inherit a rich higher-order multipolar composition from their origin as BICs, where the net ED moment vanishes[38]. The detailed analysis and comparisons of the optical modes can be referred to in our previous works[46,47].

As schematically shown in Fig. 1a, the van der Waals heterostructure made of MoS$_2$ and WSe$_2$ monolayers was transferred onto the metasurface and connected to the coated Au electrodes with low contact resistance, where the charge transfer among the MoS$_2$/WSe$_2$ interface and the WSe$_2$/Ag interface is critical to the photodetector (Fig. 1e, f). The optical micrograph is shown in the inset of Fig. 1a. $n$-doped MoS$_2$ and $p$-doped WSe$_2$ monolayers were synthesized by the chemical vapor deposition (CVD), as shown in Supplementary Fig. 7. To form van der Waals vertical heterostructures, the MoS$_2$ and WSe$_2$ monolayers were transferred onto the metasurface by the dry-transfer method with a stacking angle of 0°[17]. From the Raman scattering spectra in Fig. 1g, the wavenumber differences between $A_{1g}$ and $E^1_{2g}$ are 18.1 and 10.8 cm$^{-1}$ for MoS$_2$ and WSe$_2$, respectively, confirming their monolayer nature[50,51]. The MoS$_2$ conduction band and the WSe$_2$ valence band can constitute a type-II heterostructure, which can make the interlayer excitons of ~1.55 eV appear[17], as shown in the Fig. 1e. Interlayer excitons are revealed by the absorption spectra with 1 s

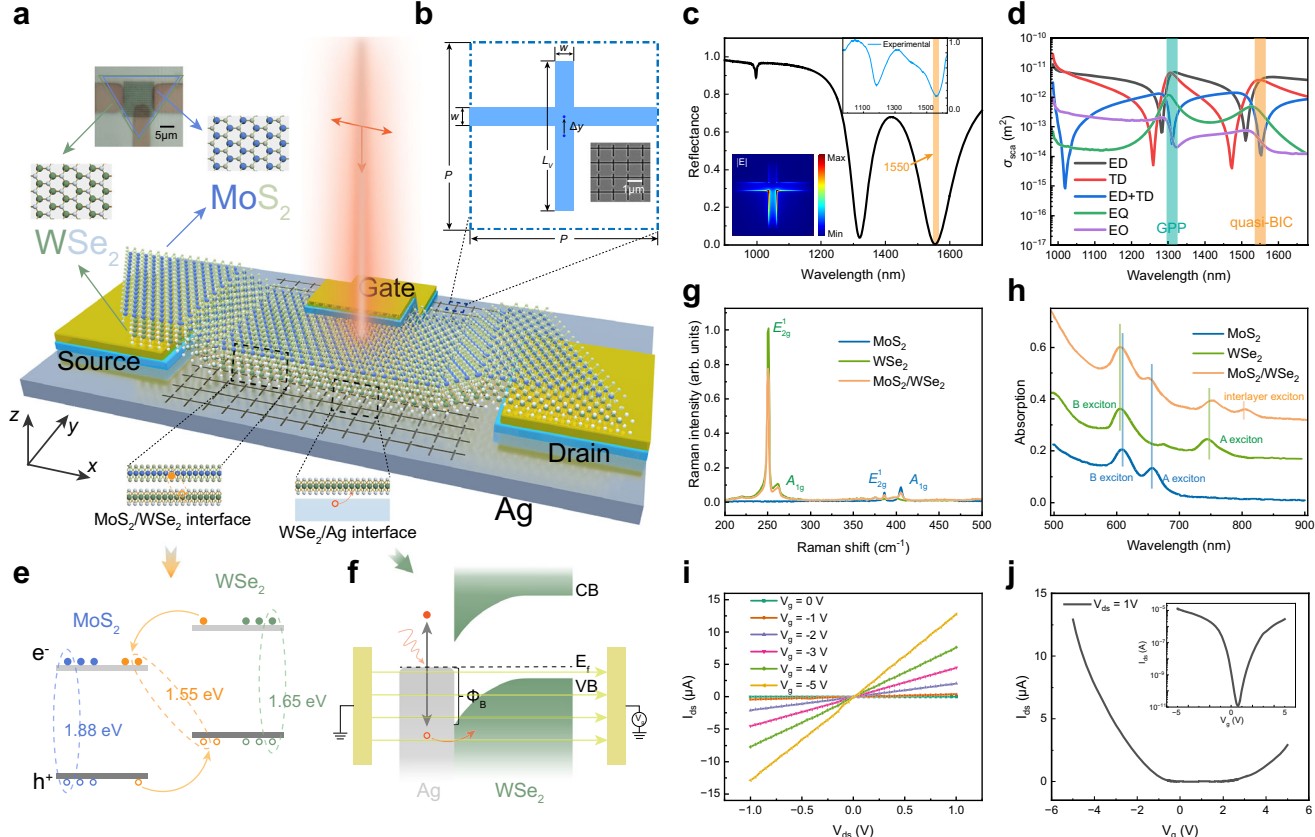

**Fig. 1 | Architecture and properties of the photodetector. a** Schematic of the photodetector based on the plasmonic metasurface loaded with MoS₂/WSe₂ heterostructure. The electrodes consist of Au (30 nm)/SiO₂ (100 nm). Inset: Optical micrograph of the device. **b** The unit cell of the metasurface, which consists of two orthogonal grooves with the same width $w = 100$ nm. The horizontal groove crosses through the period, and the vertical groove has a length of $L_V = 800$ nm. The etching depth is optimized as $d = 300$ nm, and the array period is defined as $P = 1000$ nm. The structural asymmetry is defined by shifting the horizontal groove $\Delta y$ along the $y$ direction. Inset: The SEM image of the metasurface. SEM: scanning electron microscope. **c**, The simulated and measured reflection spectrum of hybrid structure with $\Delta y = 100$ nm, with input light of horizontal polarization. The mode wavelength of the quasi-BIC is fixed at 1550 nm (orange shaded area). Inset: The calculated electric field intensity distributions of one unit on the surface. BIC: bound state in the continuum. **d**, Multipole decomposition spectra of the quasi-BIC.

ED electric dipole, TD toroidal dipole, EQ electric quadrupole, EO electric octupole. GPP gap plasmon polariton. **e** Energy level diagram of MoS₂/WSe₂ heterostructure. e⁻: electrons with a negative charge; h⁺: holes with a positive charge. The dash lines represent the binding of different excitons. **f** Diagram of hot carrier injection from single-crystalline Ag into WSe₂ at $V_g = 0$ V. VB: valence band; CB conduction band. $E_f$: Fermi level. $\Phi_B$: Schottky barrier height. The yellow arrows represent the direction of the electric field under the working gate voltage. **g** Raman spectra of isolated MoS₂, WSe₂, and MoS₂/WSe₂ heterostructure. Excitation laser: 532 nm, continuous wave. **h** Absorption spectra of MoS₂, WSe₂, and MoS₂/WSe₂ heterostructure. Vertical lines represent the position of the A, B and interlayer excitons among the heterostructure. **i** $I_{ds}$-$V_{ds}$ curves of the device at different gate voltages under dark condition. $V_{ds}$: source-drain voltage; $I_{ds}$: source-drain current. **j** Gating response of the device under dark condition at $V_{ds} = 1$ V. Inset: The diagram with logarithmic coordinates. $V_g$: gate voltage.

exciton states, as shown in Fig. 1h, in which the 610 nm peak corresponds to B excitons of WSe₂ and MoS₂ monolayers, the 655 and 750 nm peaks correspond to A excitons of MoS₂ and WSe₂ monolayers, and the 800 nm peak corresponds to the interlayer exciton of the MoS₂/WSe₂ heterostructure[29]. The red- and blue-shift behavior of absorption peaks in the heterostructure with respect to those of isolated MoS₂ and WSe₂ monolayers stems from charge transfer between MoS₂ and WSe₂, which is further corroborated by photoluminescence (PL) spectra (Supplementary Fig. 8).

The electrodes on single-crystalline Ag were prepared by lithography and magnetron sputtering of Au (30 nm)/SiO₂ (100 nm), in which the SiO₂ layer guarantees electrical insulation. The electrical contact of the device under dark condition is depicted in Fig. 1i. The current-voltage ($I_{ds}$-$V_{ds}$) characteristics exhibit a linear relationship under varying gate voltages ($V_g$), indicating Ohmic contacts between the heterostructure and electrodes. To investigate the electrostatic doping effect, $V_{ds}$ was kept at 1 V, and the top gate voltage was varied to measure $I_{ds}$, as shown in Fig. 1j. The device shows hole-dominated behavior with significant increase of $I_{ds}$ when $V_g < 0$. The gate-tunable performance of this photodetector can be explained as follows. With a

negative gate voltage, the trap states in the heterostructure are filled with holes, preventing photoinduced holes from being captured, thus engendering more holes for the photocurrent. Besides, the hot carrier injection behavior could also be tuned with the gate voltage. The Schottky barrier between WSe₂ and Ag is -1.06 eV[10,52], as shown in Fig. 1f. When a negative gate voltage is applied, the hot hole from the Ag substrate would be accelerated by the electrical field, which makes hot hole injection easier, thus contributing to the responsivity[29].

## Performance of the photoelectric enhancement

In the MoS₂/WSe₂ heterostructure, a number of excitons enable a broadband photoelectric response. When the energy of incident photons is lower than that of excitons and interlayer excitons in the MoS₂/WSe₂ heterostructure, multiphoton absorption processes, such as TPA, become the dominant excitation mechanism. As shown in Fig. 2a, electrons in the ground state (GS) can absorb two low-energy photons, transit to a higher energy state, and then relax to the 1 s exciton state. This extends the photoelectric response into the NIR-II region. Typically, the efficiency of such nonlinear processes is strongly improved by the resonances within the device, as illustrated in Fig. 2a.

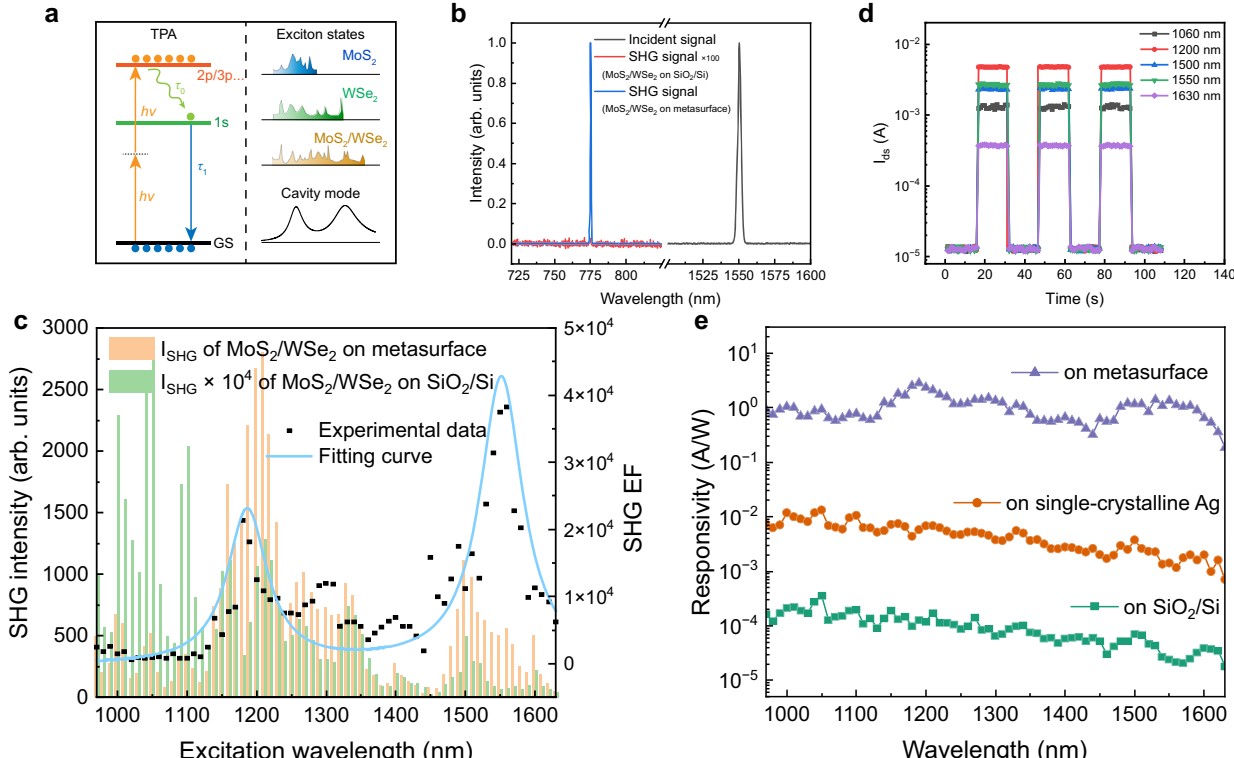

**Fig. 2 | Optical and photoelectric enhancement of the device. a** Left: Schematic of TPA processes at exciton resonance. Right: Resonances of the device. TPA: two-photon absorption. GS: ground state. $hv$: energy of a photon. $\tau_0$, $\tau_1$: relaxation and transition times. **b** Normalized spectra of fundamental laser (right, 1550 nm) and SHG signal (left, 775 nm) generated from the MoS$_2$/WSe$_2$ heterostructure on SiO$_2$/Si (magnified 100 times) and Ag metasurface. SHG: second harmonic generation. **c** Measured SHG intensity of the MoS$_2$/WSe$_2$ heterostructure on SiO$_2$/Si and Ag metasurface under excitation wavelength from 975 to 1625 nm. The signal intensity of that on SiO$_2$/Si substrate has been magnified $10^4$ times here. Right axis: Calculated SHG EF from experimental data. The curve is fitted by Lorentz fitting. EF: enhancement factor. **d** On/off switching behavior of the MoS$_2$/WSe$_2$ heterostructure on Ag metasurface at a laser excitation of 2 mW. **e** Photoresponse of the MoS$_2$/WSe$_2$ heterostructure on Ag metasurface, single-crystalline Ag, SiO$_2$/Si substrates in the NIR-II region. NIR-II: near-infrared second window.

On one hand, the optical response is enhanced through resonant absorption when the excitation wavelength aligns with exciton energy levels, such as the 2p and 3p exciton states[53]. On the other hand, the cavity mode of the Ag nanostructure also plays a pivotal role in amplifying the nonlinear process, which is the focus of our work. It is worth noting that TPA, a third-order nonlinear process, is inherently weak and often challenging to detect directly. Alternatively, second harmonic generation (SHG), a second-order nonlinear process, can serve for characterizing the analogous nonlinear behavior in the device. Although SHG is distinct from TPA in terms of physical origin and order, the efficiencies of both scale with the fourth power of the input light's electric field. Consequently, field enhancement trends of TPA and SHG are identical (see Supplementary Note 11 in supplementary information for details).

For the measurement of nonlinear optical and photoelectric processes, excitation wavelengths were generated by a tunable pulsed laser (repetition rate = 76 MHz) of 2 ps with the laser power ~2 mW and a spot size of ~1 μm on the sample. Figure 2b shows the SHG spectra of the MoS$_2$/WSe$_2$ heterostructure on (blue line) and off (red line) the metasurface, respectively, confirming an ~40000-fold enhancement with the pump laser in resonance with the quasi-BIC (1550 nm), as shown in Fig. 1c. Such large SHG enhancement is attributed to the high field enhancement of the quasi-BIC achieved in our structure. When scanning the pumping laser among the NIR-II region, there exist a series of resonance peaks, corresponding to exciton states of the heterostructure, as shown in Fig. 2c. In general, the SHG intensity on SiO$_2$/Si decreases with the wavelength, as a result of reduced involvement of excitons in the nonlinear optical process at longer excitation wavelengths. Upon integration with the metasurface, the SHG intensities of the heterostructure were significantly enhanced. When normalizing the SHG signal of the heterostructure on the metasurface to that on the SiO$_2$/Si substrate, the EFs provided by the metasurface at different wavelengths were obtained, as shown in Fig. 2c. It can be seen that, the fitted curve of the EFs demonstrates a strong correlation with the reflection spectrum presented in Fig. 1c. Note that, the metasurface itself can also generate SHG, which is significantly weak (more than five orders of magnitude weaker than that from TMDCs) and is therefore ignored in our analysis. Unlike all-dielectric metasurfaces, whose performance stems from a high Q factor, plasmonic metasurfaces achieve strong field enhancement primarily through their small mode volume. Consequently, plasmonic metasurfaces can exhibit broadband enhancement with relatively low Q factor. In addition, the single-crystalline Ag plays an important role (Supplementary Note 10 in supplementary information) here for that the performance of optical mode is largely influenced by the material loss. In our previous work, a greater SHG EF was observed at a shorter wavelength, which can be attributed to more desirable material index[47].

The photoelectric measurements of the device were operated at low working voltage: $V_{ds} = 1$ V and $V_g = -5$ V. Figure 2d illustrates the photoelectric responses of the device, with the photocurrent, $I_{phc} = I_{on} - I_{off}$, attaining values of several milliamperes in the NIR-II region. Responsivities, defined as $R = I_{phc}/P$, where $P$ denotes incident power, were calculated under various conditions to evaluate the enhancement effects of the nanostructure, as shown in Fig. 2e. For the MoS$_2$/WSe$_2$ heterostructure on SiO$_2$/Si substrate, the observed spectral peaks

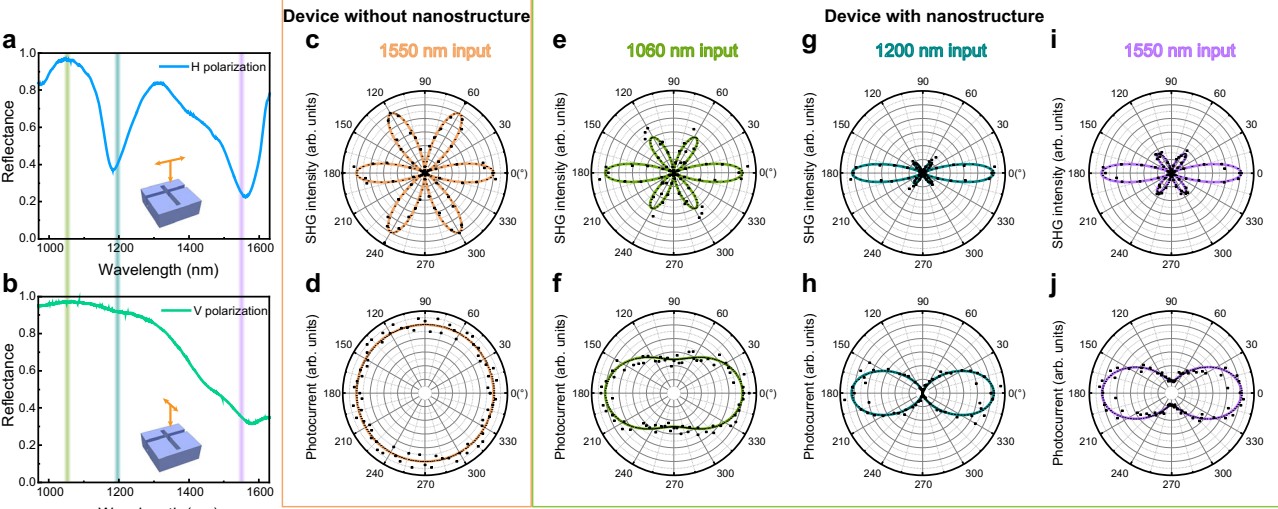

**Fig. 3 | Linear polarization characteristics.** Measured reflectance spectra of the hybrid structure with input light of horizontal (**a**) and vertical (**b**) polarization. The three colored shaded areas represent the measured wavelengths of 1060, 1200 and 1550 nm in (**c**–**j**). Under 1550 nm light illumination, polar plots of the normalized perpendicular component of SHG intensity (**c**) and polar plots of photocurrents (**d**) from the MoS$_2$/WSe$_2$ heterostructure on the SiO$_2$/Si substrate with the entire 360° rotation of the sample. Under 1060 nm (**e**, **f**), 1200 nm (**g**, **h**) and 1550 nm (**i**, **j**) light illumination, polar plots of the normalized perpendicular component of SHG intensity (**e**, **g**, **i**) and polar plots of photocurrents (**f**, **h**, **j**) from the MoS$_2$/WSe$_2$ heterostructure on the metasurface with the entire 360° rotation of the sample. The data points in **c**–**j** represent the measured data and the solid lines are the fitting curves fitted by Eq. (1) (**c**, **e**, **g**, **i**) and Eq. (2) (**d**, **f**, **h**, **j**).

closely resemble those found in SHG spectra, attributed to the exciton states. When the heterostructure is placed on a smooth single-crystalline Ag substrate, the photoelectric response exhibits an enhancement of about 58-fold. Therein, the carrier mobility of the structure on single-crystal Ag (87 cm$^2$V$^{-1}$s$^{-1}$) is approximately twice that on SiO$_2$/Si (43 cm$^2$V$^{-1}$s$^{-1}$). Meanwhile, the plasmonic surface provides a field enhancement of $E_{enhanced}/E_{input} = 2$. Then the enhancement of TPA can be calculated as $(E_{enhanced}/E_{input})^4 = 16$. Thus, the rest increase is ~1.78-fold, owing to the exciton-based interface engineering, as thoroughly investigated in our previous research[29]. The power of excitons dissipated into metal-related nonradiative channels, such as SPPs and electron-hole excitations[54,55], has possibility to be re-utilized for photoelectric devices. That is, these decay products, particularly hot holes, can be subsequently recycled into photocurrent. And the negative gate bias promotes such process, as illustrated in Fig. 1a. Furthermore, for the heterostructure on the Ag metasurface, the responsivities were improved by an additional 2–3 orders of magnitude, a result that can be ascribed to the enormous electric field enhancement induced by the metasurface structure. The responsivity reaches 1.35 A/W at 1550 nm, $5 \times 10^4$ times larger than that of MoS$_2$/WSe$_2$ heterostructures on SiO$_2$/Si substrate. Besides, the noise equivalent power (NEP) of the photodetector is measured as $8.96 \times 10^{-13}$ W/Hz$^{1/2}$, and the corresponding specific detectivity $D^*$ is calculated to be $2.08 \times 10^8$ Jones at 1550 nm (Supplementary Fig. 14). Owing to a giant photoconductive gain (Supplementary Note 16 in supplementary information), the external quantum efficiency (EQE) reaches 108% at 1550 nm.

Prominently, only about half of the photocurrent originates from direct photocarrier generation. Comparing the SHG EFs and the photocurrent EFs between MoS$_2$/WSe$_2$ heterostructure on the Ag metasurface and single-crystalline Ag, it is found that the photocurrent enhancement exceeds the purely optical SHG enhancement. In the case of SHG, the entire signal comes exclusively from the MoS$_2$/WSe$_2$ heterostructure. By contrast, the photocurrent generated by the MoS$_2$/WSe$_2$ heterostructure on the Ag metasurface arises not only from the heterostructure itself but also from hot holes injected from the Ag metasurface. Except for those dissipated from excitons, hot holes are also produced from the decaying of plasmonic modes at the excitation wavelength. The former is associated with TPA

within the MoS$_2$/WSe$_2$ heterostructure, while the latter results from intrinsic light absorption by the Ag metasurface. From Supplementary Figs. 12,13, the exciton-related (or TPA-related) proportion of photocurrent is ~90% at 1550 nm, while that involved directly generated hot holes via Ag metasurface at 1550 nm is ~10%. Independent of the semiconductor's bandgap, hot carrier injection is a linear process, making it a valuable and complementary component of the overall photoresponse, particularly under low-power excitation (e.g., under continuous-wave illumination, as shown in Supplementary Fig. 12b). Furthermore, hot carrier transfer holds the potential to overcome the intrinsic spectral limitations of semiconductors—an area that has attracted growing research interest[56]. Beyond spectral extension, metallic nanostructures can also enable multidimensional photon detection, selectively discriminating optical signals based on chirality, phase, and polarization[57,58].

## Polarization analysis of the device

In the above situations, the excitation light is polarized in the horizontal (H) orientation to match the quasi-BIC mode, the reflectance spectra of which is shown in Fig. 3a. Under vertical (V) polarization, a GPP mode is also supported, primarily confined within the horizontal groove of the structure, as shown in Fig. 3b. The different optical resonances arising from varying input light polarizations result in linear polarization-resolved optical and photoelectric responses. For the MoS$_2$/WSe$_2$ heterostructure on the SiO$_2$/Si substrate, the polarization characteristics of the SHG signal are also shown in Fig. 3c. With an H-polarized input laser, V-polarized SHG signal is collected when the azimuth angle of the sample (see the optical micrograph shown in Fig. 1a) rotates from 0° to 360°, and a typical petal pattern with 6-fold symmetry is obtained[59]. For a stacking angle of 0° between MoS$_2$ and WSe$_2$ monolayers, the polarization-resolved response of the heterostructure owns the same characteristics as single TMDCs[60]. Fig. 3d shows the corresponding photocurrent under different excitation polarizations (the same photoelectric test condition as that in Fig. 2). The photocurrent remains isotropic with respect to polarization angle, as it is primarily determined by the material itself. When placed onto the metasurface, the polarization-resolved optical and photoelectric responses are largely influenced by the optical mode within the

metasurface, as shown in Fig. 3e–j. For the SHG signal, the data can be fitted by the equation:

$$I_{SHG\perp}(\theta) = [I_{SHG_{min}}^{1/2}\sin^2(\theta+\varphi) + I_{SHG_{max}}^{1/2}\cos^2(\theta+\varphi)]^2 \times \cos^2 3\theta, \quad (1)$$

where $I_{SHG\ min}$, $I_{SHG\ max}$ reflect the minimum and maximum SHG intensity, $\theta$ is the angle between the polarization and the edge of TMDCs, and $\varphi$ is the angle between the edge of TMDCs and the $x$-direction of the structure, which is equal to zero here. Similarly, for the photocurrent, the data can be fitted by the equation:

$$I_{phc}(\theta) = [I_{phc\ min}^{1/2}\sin^2(\theta+\varphi) + I_{phc\ max}^{1/2}\cos^2(\theta+\varphi)]^2, \quad (2)$$

where $I_{phc\ min}$, $I_{phc\ max}$ reflect the minimum and maximum photo-current. For the SHG component parallel to the excitation light, the term $\cos^2 3\theta$ in Eq. (1) is replaced by $\sin^2 3\theta$[59,60], as shown in Supplementary Fig. 17. Then, the total SHG signal $I_{SHG} = I_{SHG\ \perp} + I_{SHG\ \parallel}$ can be written the same as Eq. (2). Hence, the polar plots of SHG intensity and photocurrent mutually corroborate the device's capability for polarization resolution.

In the case of the MoS$_2$/WSe$_2$ heterostructure on the SiO$_2$/Si substrate, $I_{phc\ min} = I_{phc\ max}$, and the polarization ratio (defined as $I_{phc\ max}/I_{phc\ min}$), PR = 1. A larger PR indicates a better ability for polarization resolution. At the wavelength of 1200 nm, as shown in Fig. 3g, h, the PR attains a value of up to 40, representing a notably high performance compared to other photodetectors based on 2D materials, which is attributed to the different performance between the optical mode excited by H- and V-polarized light in the metasurface. As mentioned above, there is a GPP mode at 1200 nm under H-polarization, as shown in Fig. 3a. While no resonance exists nearby under V-polarization, as shown in Fig. 3b. Similarly, the PRs of varying wavelength are dependent on the mode resonance under different polarizations. At the wavelength of 1060 nm, where no resonance exists under both polarizations, PR = 2. At the wavelength of 1550 nm, PR = 6 proves that the quasi-BIC under H-polarization exhibits greater enhancement than the GPP mode under V-polarization. In fact, the mode wavelength of the metasurface under V-polarization can be regulated independently by the length of the horizontal groove, as shown in Supplementary Fig. 18. Then, the PR of different wavelengths can be modulated manually, which expands the potential of our device for constructing linear-polarization-sensitive photodetectors.

## Design of chirality-resolved photodetector

Beyond linear polarization control, we further extended the functionality to chiral light detection by breaking the mirror symmetry of the metasurface along the $x$-direction. To introduce in-plane chirality, the horizontal groove length ($L_H$) was designed to be finite. To maintain the target operational wavelength, the period was increased to $P = 1200$ nm, while all other parameters remained consistent with the structure in Fig. 1a. As shown in Fig. 4a, under circularly polarized excitation, the structure with $\Delta x = 0$ exhibits identical optical responses for left- and right-handed circular polarizations (LCP and RCP), due to the preserved mirror symmetry. Two resonant modes are observed: one associated with the quasi-BIC (blue dashed line), excited predominantly by the horizontal field component of the incident light, and another corresponding to the GPP mode (green dashed line) supported by the horizontal groove, driven by the vertical field component. The maximum circular polarization response occurs at the spectral overlap of these two modes. Introducing a lateral shift of $\Delta x = 50$ nm in the vertical groove (inset, Fig. 4b) breaks the mirror symmetry along the $x$-direction. Figure 4b shows the mapped reflection spectra under LCP illumination for varying $\Delta y$s. For $\Delta y > 0$, the hybridized behavior of the horizontal and vertical modes remains similar to the symmetric case. In contrast, when $\Delta y < 0$, the splitting occurs between the two modes, strongly suppressing the response to

LCP light. Under RCP illumination, the mapped reflection spectra exhibit similar but mirror-symmetrical characteristics to that of LCP illumination, as shown in Fig. 4c. Consequently, this tailored asymmetry enables pronounced chirality-specific responses, allowing the device to distinguish between LCP and RCP without external optical elements. For the multipole decomposition after breaking the symmetry along the $x$-direction, it does not destroy the anapole state but rather makes it chiral-dependent, as shown in Supplementary Fig. 23.

Experimentally, structures with $\Delta x = 50$ nm and varying $\Delta y$s are fabricated by FIB. Figure 4d, e show the measured reflection spectra under different circularly polarized light, with the SEM images by the side. Under the LCP illumination, the mode resonances demonstrate a combination of both horizontally and vertically polarized modes, marked by the orange line, as shown in Fig. 4d. While for the RCP light, the mode resonances predominantly exhibit characteristics corresponding to the horizontally polarized mode, marked by the blue line, as shown in Fig. 4e. And the vertically polarized mode marked by green line is covered up by the horizontally polarized one. When $\Delta y = 250$ nm, the electric field intensity distributions at the wavelength of 1550 nm are shown in Fig. 4f. It can be observed that the fields are distributed within both the vertical and horizontal grooves under the LCP illumination, corresponding to the horizontally polarized quasi-BIC mode and the vertically polarized GPP mode, respectively. The field intensity associated with the former is greater than that of the latter, attributable to the inherent characteristics of quasi-BICs. For RCP light, the field intensity mainly exists among the horizontal groove, corresponding to the vertically polarized GPP mode. Then, the performance of the metasurface for the LCP light is much better than that for the RCP light, which brings chirality-resolved optical and photoelectric responses.

Figure 4g shows the reflection and the SHG spectra of the MoS$_2$/WSe$_2$ heterostructure on the metasurface with $\Delta x = 50$ nm and $\Delta y = 250$ nm under LCP (orange) and RCP (blue) illuminations (the same photoelectric test condition as that in Fig. 2). The degree of circular polarization (DCP) of SHG emission, defined as $|I_{SHG\ LCP} - I_{SHG\ RCP}| / |I_{SHG\ LCP} + I_{SHG\ RCP}|$, reaches a value of 0.804 at the wavelength of 1550 nm. In parallel, the photoelectric response exhibits a corresponding chiral behavior, as illustrated in Fig. 4h. The responsivity spectra measured at the device (Fig. 4h, top) are consistent with the reflectivity spectra (Fig. 4g, top). The plasmonic resonance engenders the field enhancement and thus the responsivity. The chiral discrimination ratio, defined as $I_{phc\ LCP}/I_{phc\ RCP}$, of the photoelectric response can reach up to 7.2 at the wavelength of 1550 nm, as shown in the bottom of Fig. 4h. The spectrum of discrimination ratio forms a peak around the resonance of the LCP-optical mode. The responsivity was also measured as a function of the light ellipticity at the wavelength of 1550 nm, as shown in Fig. 4i. It reaches a maximum at 45° or 225° (LCP) and a minimum at 135° or 315° (RCP). Notably, the responsivity under the horizontally polarized illumination (0° or 180°) is larger than that under the vertically polarized illumination (90° or 270°). That is because the horizontally polarized quasi-BIC mode is superior to that of vertically polarized GPP mode. Besides, the characteristics of the device under LCP and RCP illuminations can swap with each other when setting the $\Delta y = -250$ nm, as shown in Fig. 4b, c.

In summary, we have demonstrated a high-performance, multi-functional photodetector by integrating a MoS$_2$/WSe$_2$ van der Waals heterostructure with an anapole-inspired metasurface (Supplementary Table 1). This architecture successfully overcomes the intrinsic limitations of TMDC-based devices, including weak light absorption and restricted spectral response. Through the excitation of quasi-BICs and anapole states, the metasurface achieves tremendous localized field enhancement in virtue of the high-order multipoles, significantly boosting the TPA efficiency and enabling broadband photodetection deep into the NIR-II window. Moreover, the decay of excitons and plasmonic resonances facilitates efficient hot-carrier injection,

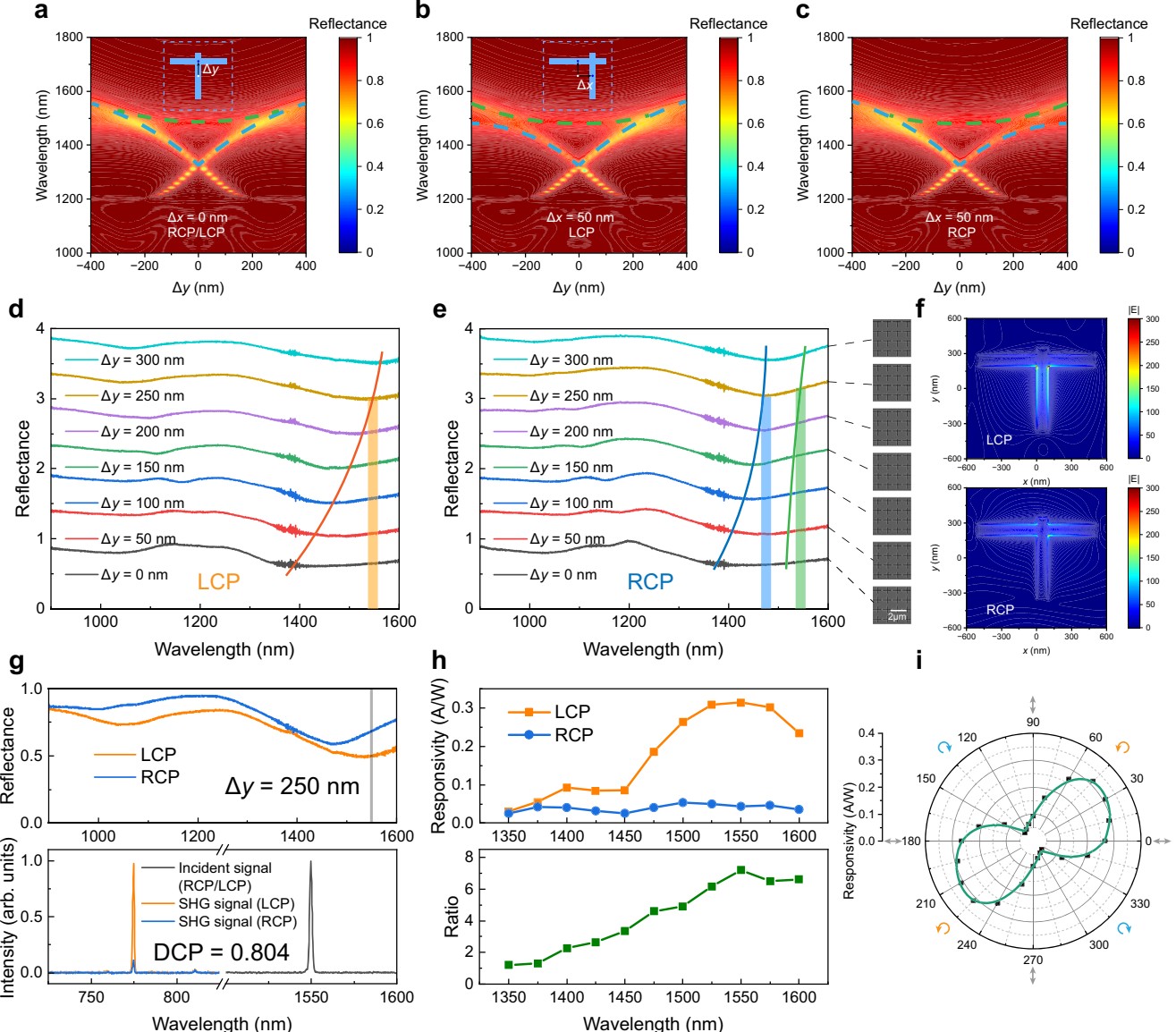

**Fig. 4 | Chiral photoelectric response. a** Calculated mapping image of reflection spectra under LCP and RCP illumination with structural asymmetry $\Delta y$. The period is enlarged to $P = 1200$ nm compared to the structure in Fig. 1. LCP left-handed circular polarizations, RCP right-handed circular polarizations. The blue and the green dashed lines represent the optical resonances excited by the horizontal and vertical field component of the incident light, respectively. Calculated mapping image of reflection spectra with $\Delta y$ under LCP (**b**) and RCP (**c**) illumination. Mirror symmetry is broken by shifting the horizontal groove along $x$ direction, $\Delta x = 50$ nm. Measured reflection spectra with $\Delta y$ under LCP (**d**) and RCP (**e**) illumination, $\Delta x = 50$ nm. The images of SEM are listed by the side. **f** The calculated electric field

intensity distributions of one unit on the surface of metasurface for the LCP and RCP, at the wavelength of 1550 nm, $\Delta x = 50$ nm and $\Delta y = 250$ nm. **g** Top: Measured reflection spectra with $\Delta x = 50$ nm and $\Delta y = 250$ nm under LCP and RCP illuminations. Bottom: Normalized spectra of fundamental laser (right, 1550 nm) and SHG signal (left, 775 nm) generated from the MoS$_2$/WSe$_2$ heterostructure on Ag metasurface under LCP (orange) and RCP (blue) illumination. **h** Top: Spectra of responsivity induced by LCP and RCP light. Bottom: The chiral discrimination ratio of the photoelectric response. **i** Ellipticity angle dependent responsivity polar diagram at the wavelengths of 1550 nm. The data points represent the measured data and the solid line is the smooth connection of the data points.

markedly improving photocurrent generation and overall quantum efficiency. The resulting device exhibits a remarkable responsivity of 1.35 A/W at 1550 nm while operating at room temperature. Furthermore, by breaking the in-plane symmetry of the metasurface, the capability of chiral discrimination is successfully incorporated. This work not only provides a viable pathway toward ultracompact, broadband, highly sensitive photodetectors with built-in polarization functionality but also establishes a general paradigm of leveraging nanophotonic engineering to manipulate light–matter interactions in low-dimensional semiconductors. This technology is anticipated to open new avenues for applications in NIR-II bio-imaging, high-speed

optical communication, portable spectroscopic sensors, and flexible optoelectronic systems.

## Methods

### 2D material preparation

Monolayer MoS$_2$ and WSe$_2$ were prepared using the CVD method. All powders were purchased from Aladdin. S powder weighted 60 mg was placed in a ceramic boat. A mass ratio of MoO$_3$ to NaCl of 2:1 was maintained. The two powders were then uniformly mixed and placed in another ceramic boat as precursors. SiO$_2$/Si substrates sized $5 \times 3$ cm$^2$ were placed on top of the ceramic boat, facing downwards

the mixed powders. The ceramic boat was then placed in the heating center of the furnace, with an 18 cm distance between the two ceramic boats. Ar flow was maintained at 80 sccm and the temperature of the heating center was set to 840 °C. The system was kept at this temperature for 4 min before cooling to room temperature. As to monolayer $WSe_2$, the preparation process was similar to that of $MoS_2$. The distance between the Se powder (60 mg) and the $WO_3$/NaCl mixture (mass ratio 2:1) was kept at 16 cm. Ar and $H_2$ flows were maintained at 80 sccm and 5 sccm, respectively, to convey Se vapor to the $MoO_3$ source. The synthesis was conducted at 850 °C for 4 min. Both monolayer $MoS_2$ and $WSe_2$ were prepared in a furnace under standard atmospheric pressure.

## Synthesis of single-crystalline Ag

The single-crystalline Ag was synthesized by a polyol reduction method catalyzed by Pt nanoparticles and controlled by $NH_4OH$. The steps are as follows. 2.5 mmol $AgNO_3$ (HUSHI) and 15 ml ethylene glycol solution (AR, HUSHI) as precursors were poured into a glass conical flask. 3 ml $NH_3OH$ (25–28%, HUSHI) was added into it. Magnetic stirring was used to mix when 0.46 g polyvinylpyrrolidone (Mw = 55,000, Sigma-Aldrich) was added. 8 μl $H_2PtCl_6$ (8% in $H_2O$, Aladdin) was added and 1.8 ml $H_2O_2$ (30%, HUSHI) was injected into the solution to reduce Ag. Single-crystalline Ag was then synthesized after 5 days' reaction at room temperature. The Ag crystals were purged by deionized water, acetone, and ethyl alcohol respectively, and preserved in alcohol.

## Numerical calculations

The finite-difference time-domain method (Lumerical FDTD Solutions, Ansys, Inc.) and finite-element method (Comsol Multiphysics, COMSOL Co. Ltd.) were mainly used for numerical calculations. The optical parameters from Johnson and Christy[61] were utilized for simulations. For accuracy, the finest mesh size is set no more than 2 nm. TM-polarized (electric field direction is parallel to the $x$-axis) plane wave was incident perpendicularly to the structure. Theoretical SHG and TPA EF is defined as EF = average $|\boldsymbol{E}|^4/|\boldsymbol{E}_0|^4$, where $\boldsymbol{E}_0$ is the electric field of incident light, and $\boldsymbol{E}$ is the enhanced electric field among the structure.

## The fabrication of plasmonic structures

Experimentally, we used FIB (Helios 600, FEI, Inc.) etching to gain the required nanostructures. The ion source is $Ga^+$, the accelerating voltage is 30 keV, and the ion beam current is 7.7 pA. The dose used in etching is 1.13 $μm^3$/nC. The achievable linewidth of $Ga^+$ FIB etching is about 10–15 nm, which is sufficient for the structures we require.

## Optical and electrical measurements

Optical measurements were performed using a confocal microscope (Horiba). For broadband photodetection, a tunable laser (Coherent), comprising a pulsed laser (Mira 900-D, 2 ps), an optical parametric oscillator, and second harmonic generation, was integrated with the confocal microscope. The tunable laser emits picosecond pulses of light ranging from 350 nm to 1650 nm at a repetition rate of 76 MHz. The photoelectric response signal was collected by a Keithley 4200 system. All optical and electrical measurements were conducted at room temperature and under atmospheric conditions.

## Data availability

Relevant data supporting the key findings of this study are available within the article and the Supplementary Information file. All raw data generated during the current study are available from the corresponding authors upon request.

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

## Acknowledgements

This work was supported by the National Key R&D Program of China (2022YFA1405004 (C.Z. and Y.L.), 2017YFA0303700 (X.Z., C.Z., and Y.L.), and 2023YFA1406603 (J.D.)), the National Natural Science Foundation of China (11274159 (X.Z.), 11374150 (Y.Z.), T2394473 (J.D.), and 12374112 (J.D.)), and the Natural Science Foundation of Jiangsu Province, Major Project (BK20212004 (Y.L.)).

## Author contributions

Q.Z., Z.D., S.F., and X.H. conducted the experiments. Q.Z. and K.L. performed the simulations and theoretical analyses. Q.Z. and Z.D. conceived the idea. Y.-L.C., C.Z., J.D., Y.L., Y.Z., Y.-F.C., and X.Z. supervised the project. Q.Z., Z.D., K.L., and X.Z. wrote the manuscript with inputs from all authors. All authors participated in analyses and discussions.

## Competing interests

The authors declare no competing interests.
