## [Transparent Peer Review file · Nature Communications]

Anapole-state-enhanced 2D chiral photodetector operating in the near-infrared second window

Corresponding Author: Professor Xuejin Zhang

Version 0:

Reviewer comments:

Reviewer #1

(Remarks to the Author)

Comments to the Authors

The manuscript presents a MoS₂/WSe₂ heterostructure photodetector integrated with a plasmonic metasurface in the NIR-II. The authors use anapole states and quasi-bound states in the continuum (quasi-BICs) to realize significant electromagnetic field enhancement, enabling efficient two-photon absorption (TPA) and hot-carrier injection. The resulting responsivity is 1.35 A/W at 1550 nm. Furthermore, chiral discrimination is demonstrated by the engineered asymmetry of the metasurface. The manuscript, in its current form, does not meet the high standards for originality and comprehensive analysis required for publication in Nature Communications. The key novelty of this study may reside in the realization of a multifunctional photoresponse (polarization sensitivity and chiral discrimination) via MoS₂/WSe₂ heterostructure, given the strong technological shift toward multifunctional detection modalities beyond light intensity. Therefore, the authors are required to fully address the ensuing questions and comments to substantially strengthen the core claims and provide sufficient justification for publication.

Detailed remarks:

1. Photoresponse mechanism of two-photon absorption (TPA) and hot-carrier injection

The reliance on SHG measurements to characterize TPA enhancement is noted (Line 155). The claim that 90% of the photocurrent at 1550 nm is attributed to TPA (Line 203 and Fig. S10) is critical but needs clearer substantiation.

Although Fig. S9 presents power-dependent optical and photoelectric signals, the main text lacks a clear, quantitative description of this relationship. The authors must fit the power-dependent photocurrent data using the relation $I = k \cdot P^\alpha$ and explicitly state the fitted exponent α in the main text.

This is a better method for clarifying the TPA photocarrier generation mechanism when $\alpha \approx 2$. A fitted exponent of $\alpha \approx 1$ suggests that the plasmonic resonance-induced hot-carrier mechanism is predominant. This fitting is essential to support the claims regarding TPA dominance. Please also include the measured TPA coefficient (β) of the device.

2. Photodetector figure of merit

The high responsivity is achieved under external bias, which invariably increases the dark current and noise. Performance metrics must account for this trade-off. The authors must perform noise measurements and a detectivity assessment. Please report the noise level (e.g., noise equivalent power, NEP) and the corresponding specific detectivity (D^*) and EQE.

The external bias introduces photoconductive gain. Please discuss the gain mechanism and report the measured responsivity bandwidth (speed) of the device, as high gain often trades off with speed.

3. The claim that the photonic mode employed in this work is an “anapole state” is questionable.

The analysis or discussion about this claim is not solid. “Nonradiative” is not an exclusive characteristic of an anapole state. In fact, the photonic mode employed in this work seems like a typical light funneling mode (or a plasmonic cavity mode), which has been explicitly studied. Please check the following papers about this mode: PRL 107, 093902 (2011), Nat. Commun. 7, 11283 (2016). If the authors want to keep the claim about “anapole state”, please provide more substantial analysis.

4. Relationship between “quasi-BIC” and “anapole state”

The physical mechanism explanation must be significantly strengthened by clarifying the interplay between the quasi-BIC mode and the anapole state. Please explicitly define the causal relationship (or parallel relationship) between the quasi-BIC

and the anapole state. Is the quasi-BIC a necessary consequence of the anapole-driven dominance of higher-order multipoles?

The authors should clearly describe how the perfect BIC mode is identified before symmetry breaking (δ_y). A supplementary analysis showing the asymmetry parameter (δ_y) dependent Q-factor distribution and/or the far-field polarization singularity would be highly beneficial in validating the claimed perfect BIC and the evolution of quasi-BIC. What key benefits does the anapole state offer over other high-Q photonic modes (e.g., guided mode resonance, chiral BIC) in the context of boosting photodetection efficiency?

5. Figure clarity

Figure 1a contains too many sub-panels. The authors should re-organize or re-design this figure to ensure that all labels are clear, legible, and that accurate, unambiguous citation of individual panels in the main text is feasible.

6. Fabrication and device structure description

Clarification on the device architecture, particularly regarding the insulation layers, is needed to validate the operational mechanisms. In Supplementary Information, Section 1, the text mentions: "To ensure the pad film continuity between top of single-crystalline Ag and SiO₂/Si substrate, the edges of the single crystalline Ag were cut by FIB and covered by a thick SiO₂ film of 200 nm. The other SiO₂ film with thickness of 100 nm positioned between Au substrate and Au electrodes acted as an insulation layer." Please clarify the position and function of the "Au substrate." Is the blue layer in the schematics (e.g., Fig. 1a) the SiO₂ film? How was this 100 nm SiO₂ film fabricated?

Could the authors please clarify the exact location of the "thick SiO₂ film of 200 nm" within the device structure? The authors state this SiO₂ only separates the Au contact and Ag metasurface. If the MoS₂/WSe₂ heterostructure is also separated from the metasurface (Ag) by this SiO₂ layer, the contribution of hot-carrier injection from Ag would be fully excluded. Could the authors confirm the presence of any silver oxide layer (e.g., Ag₂O) on the surface of the single-crystalline Ag metasurface?

7. References

There are many previous works about using metallic nanostructures to enhance hot-electron photoresponse. Some recent works include InfoMat.6, e12556 (2024), Nat. Electron. 7, 1004 (2024), Light Sci. Appl. 12, 176 (2023)... Please pay attention to the previous works in this field and make some discussion.

8. Typo

In Equation (2), the term "lphc msx" should be corrected to "lphc max".

Reviewer #2

(Remarks to the Author)

Reviewer #3

(Remarks to the Author)

In this manuscript, the authors successfully leverage quasi-BIC and higher-order multipoles to achieve exceptional electric field enhancement exceeding one thousand times. This work represents significant progress toward miniaturized multifunctional photodetectors for biological imaging and communication applications. However, several key issues as follow require clarification before the manuscript can be fully endorsed for publication.

(1) The authors claim that the field enhancement trends for SHG and TPA are identical due to their mutual dependence on the fourth power of the electric field. However, SHG represents a coherent process dependent on phase matching, while TPA constitutes an incoherent absorption process. Additionally, given their distinct nonlinear susceptibilities, it remains unclear whether assuming identical enhancement trends is fully justified, particularly in a system containing multiple excitonic and plasmonic resonances?

(2) The approximately 58-fold photoresponse enhancement on flat silver is explained as the product of increased carrier mobility, field enhancement for two-photon absorption, and an exciton-recycling factor. The theoretical basis for treating these contributions spanning both linear and nonlinear processes as multiplicative requires further justification. Additional modeling or controlled experiments would help decouple these mechanisms and validate the proposed model.

(3) While the authors note using a 2 ps pulsed laser for nonlinear measurements, the repetition rate remains unspecified. Providing the repetition rate and estimating the peak power density at the sample would enable accurate quantification of nonlinear efficiencies and facilitate comparisons with other studies.

(4) Given the exceptionally high reported enhancement factors, information concerning the corresponding signal-to-noise ratios and device stability under continuous or pulsed operation would be valuable. The authors should also comment on whether any saturation or degradation effects were observed at operational power levels?

(5) Further elaboration on the energy alignment between plasmon-generated hot carriers and the band structure of the heterostructure would enhance understanding of the underlying mechanisms. Specifically, how the Schottky barrier at the WSe₂ and silver interface influences hot carrier injection efficiency under different gate voltages warrants detailed discussion?

(6) Although the quasi-BIC provides a high quality factor and strong field confinement, its narrowband nature raises questions about its impact on broadband photodetection performance. The authors should address whether there exists a design trade-off between resonance sharpness and operational bandwidth?

(7) The critical importance of using single-crystalline silver to the observed performance requires further examination. Have

simulations or experimental comparisons been conducted with polycrystalline silver or other plasmonic materials to evaluate the role of material loss?

Reviewer #4

(Remarks to the Author)

The authors reported a surface-enhanced 2D TMDC photodetector by means of high-order multipoles with anapole states, as well as quasi-bound states in the continuum, to operate efficiently in the near-infrared second (NIR-II) window at room temperature. Both high-sensitive and chiral discrimination are achieved. The experiments and results are interesting. However, there are some concerns that should be further clarified.

1. An objective and comprehensive detection performance comparison is necessary as the authors have emphasized the advantages of the performance of their devices and it is also important to verify the advantages of anapole enhancement, especially comparing with other plasmonic structures.
2. Why the metal Ag was selected to fabricate the nanostructure? The metallic silver seems to be not very stable and is prone to oxidation. Will this affect the performance of the device?
3. Considering the TAP dominates the photocurrent enhancement, and its efficiency scales with the fourth power of the input light's electric field, can the TAP process still be generated if the continuous wave (CW) light source is used for excitation, but not the pulsed laser used in present results? This point will affect the practicality and universality of this work.
4. The author compares the SHG signals of heterostructure on SiO₂/Si and on metasurface. Is it possible to generate SHG signal by the metasurface itself?
5. The chiral-resolved photoresponse is achieved by further breaking the symmetry of the metasurface. Does it still support the anapole state after its symmetry is broken?

Version 1:

Reviewer comments:

Reviewer #1

(Remarks to the Author)

All my concerns have been addressed. The manuscript can be accepted for publication.

Reviewer #2

(Remarks to the Author)

Reviewer #3

(Remarks to the Author)

The authors have thoughtfully and thoroughly addressed all points raised in my previous review. Their responses are clear, well-supported by additional experimental data and theoretical analysis, and effectively enhance the manuscript's discussion of the underlying physical mechanisms, device performance metrics, and experimental methodology. I am satisfied with the revisions and recommend to acceptance it in current form.

Reviewer #4

(Remarks to the Author)

In this round of revision, the authors have addressed part of my concerns. However, several issues still remain and require further clarification or validation by the authors, for example, the first and third points.

For the first comment, I suggest that the authors include a table offering a comprehensive and thorough comparison of key performance metrics among representative near-infrared photodetectors.

For the third comment, I believe the authors should further evaluate and characterize the device performance using a continuous-wave (CW) light source. If the responsivity cannot reach the level obtained under pulsed illumination, this limitation should be clearly clarified and discussed in the manuscript. This is because, for photodetector applications, illumination is predominantly continuous rather than pulsed in most practical scenarios.

Response Letter to Reviewers

"Anapole-Inspired NIR-II Photodetection with High Sensitivity and Chiral Discrimination"

Research Article, NCOMMS-25-80799

A list of changes is as below:

Main text:

- (1) In the revised version, **Fig. 1a** is re-designed and redrawn. The sub-panels are renamed as **a.1, a.2, a.3, a.4**, and the related descriptions in the main text are revised, respectively.
- (2) In the revised version, **Fig. 1c** is recalculated and redrawn in the logarithmic coordinate system.
- (3) In the revised version, the words “**(Fig. S5)**” are added to the Line 8, Paragraph 1 in the “Basic structure and properties of the nanostructured device” part of **main text**.
- (4) In the revised version, the words “**(see Fig. S6 for details)**” are added to the Line 21, Paragraph 1 in the “Basic structure and properties of the nanostructured device” part of **main text**.
- (5) In the revised version, a sentence “**The detailed analysis and comparisons of the optical modes can be referred to in our previous works.^{46,47}**” is added to the ending of the Paragraph 1 in the “Basic structure and properties of the nanostructured device” part of **main text**.
- (6) In the revised version, the words “**(see Section 11 in supplementary information for details)**” are added to the ending of the Paragraph 1 in the “Performance of the photoelectric enhancement” part of **main text**.
- (7) In the revised version, the words “**(repetition rate = 76 MHz)**” are added to the Line 2, Paragraph 2 in the “Performance of the photoelectric enhancement” part of **main text**.
- (8) In the revised version, a sentence “**Note that, the metasurface itself can also generate SHG, which is significantly weak (more than five orders of magnitude weaker than that from TMDCs) and is therefore ignored in our analysis.**” is added to the Line 12, Paragraph 2 of the “Performance of the photoelectric enhancement” part of **main text**.
- (9) In the revised version, the sentences “**Unlike all-dielectric metasurfaces.....exhibit broadband enhancement with relatively low Q factor.**” are added to the Line 13, Paragraph 2 in the “Performance of the photoelectric enhancement” part of **main text**.
- (10) In the revised version, the words “**(Section 10 in supplementary information)**” are added to the Line 16, Paragraph 2 in the “Performance of the photoelectric enhancement” part of **main text**.
- (11) In the revised version, the sentences “**Besides, the noise equivalent power.....the external quantum efficiency (EQE) reaches 108% at 1550 nm.**” are added to the ending of the Paragraph 3 in the

“Performance of the photoelectric enhancement” part of **main text**.

- (12) In the revised version, the sentences “Independent of the semiconductor's bandgap, hot carrier injection.....selectively discriminating optical signals based on chirality, phase, and polarization.^{57,58}” are added to the ending of the “Performance of the photoelectric enhancement” part of **main text**.
- (13) In the revised version, the “ $I_{\text{phc msx}}$ ” is corrected to “ $I_{\text{phc max}}$ ” in Eq. (2).
- (14) In the revised version, a sentence “For the multipole decomposition after breaking the symmetry along the x -direction, it does not destroy the anapole state but rather makes it chiral-dependent, as shown in Fig. S23.” is added to the ending of the Paragraph 1 in the “Design of chirality-resolved photodetector” part of **main text**.
- (15) In the revised version, the Refs. 25–27 in the response letter are added to the main text as Refs. 56–58.
- (16) The ordinal numbers of the Figures and References mentioned in the **main text** are adjusted to the revised version.

Supplementary information:

- (17) In the revised version, the sentence “The other SiO₂ film with thickness of 100 nm positioned between Au substrate and Au electrodes acted as an insulation layer.” is replaced by “The other SiO₂ film with thickness of 100 nm positioned between Ag substrate and Au electrodes acted as an insulation layer.” in the Section 1 of supplementary information.
- (18) In the revised version, Figure S1c is redrawn as shown in Fig. R10, where the locations of “SiO₂ step of 200 nm” and “SiO₂ insulation layer of 100 nm” are labeled.
- (19) In the revised version, a paragraph “For the stability of the single-crystalline Ag.....over one month without performance degradation.” is added to the Section 1 of supplementary information.
- (20) In the revised version, Fig. R8 and the related discussion are added as the Section 5 of supplementary information.
- (21) In the revised version, Fig. R6 and the related discussion are added as the Section 6 of supplementary information.
- (22) In the revised version, the comparisons between polycrystalline and single-crystalline Ag and relative discussions (including Fig. R15d) are added as the Section 10 of supplementary information.
- (23) In the revised version, the discussions about the consistency of the enhancement of SHG and TPA are added as the Section 11 of supplementary information.
- (24) In the revised version, the fitting of the power-dependent photocurrent data and related discussions are supplied to the Section 13 of supplementary information (Section 9 of the previous version).
- (25) In the revised version, a sentence “Besides, the measured data begins.....the onset of device saturation.” is added to the ending of Section 13 of supplementary information.
- (26) In the revised version, the noise measurement and detectivity assessment (including Fig. R3) are added as the Section 15 of supplementary information.
- (27) In the revised version, the discussion of the gain mechanism, together with the responsivity speed (including Fig. R4), are added as the Section 16 of supplementary information.
- (28) In the revised version, the modulation of Schottky barrier by gate voltage and relative discussions (including Fig. R14) are added as the Section 17 of supplementary information.
- (29) In the revised version, the multipole decomposition of the chiral-resolved structures and the related discussion (including Fig. R18) are added as the Section 24 of supplementary information.

- (30) In the revised version, the Refs. 4–7,12,14–18 are added to supplementary information.
- (31) The ordinal numbers of the Figures and References mentioned in the supplementary information are adjusted to the revised version.

The main revisions are marked up with red text in the revised manuscript. The point-to-point replies to the reviewers' comments (in blue colors) are attached below.

Thank you for further consideration.

Sincerely,

Xuejin Zhang

To Reviewer #1:

Comments:

The manuscript presents a MoS₂/WSe₂ heterostructure photodetector integrated with a plasmonic metasurface in the NIR-II. The authors use anapole states and quasi-bound states in the continuum (quasi-BICs) to realize significant electromagnetic field enhancement, enabling efficient two-photon absorption (TPA) and hot-carrier injection. The resulting responsivity is 1.35 A/W at 1550 nm. Furthermore, chiral discrimination is demonstrated by the engineered asymmetry of the metasurface.

The manuscript, in its current form, does not meet the high standards for originality and comprehensive analysis required for publication in Nature Communications. The key novelty of this study may reside in the realization of a multifunctional photoresponse (polarization sensitivity and chiral discrimination) via MoS₂/WSe₂ heterostructure, given the strong technological shift toward multifunctional detection modalities beyond light intensity. Therefore, the authors are required to fully address the ensuing questions and comments to substantially strengthen the core claims and provide sufficient justification for publication.

Reply:

Thank you for your valuable time to review our manuscript. The valuable suggestions have helped us clarify the study. We have carefully studied all the comments and addressed them accordingly in the revision.

1. Photoresponse mechanism of two-photon absorption (TPA) and hot-carrier injection

The reliance on SHG measurements to characterize TPA enhancement is noted (Line 155). The claim that 90% of the photocurrent at 1550 nm is attributed to TPA (Line 203 and Fig. S10) is critical but needs clearer substantiation.

Although Fig. S9 presents power-dependent optical and photoelectric signals, the main text lacks a clear, quantitative description of this relationship. The authors must fit the power-dependent photocurrent data using the relation $I = k \cdot P^\alpha$ and explicitly state the fitted exponent α in the main text.

This is a better method for clarifying the TPA photocarrier generation mechanism when $\alpha \approx 2$. A fitted exponent of $\alpha \approx 1$ suggests that the plasmonic resonance-induced hot-carrier mechanism is predominant. This fitting is essential to support the claims regarding TPA dominance. Please also include the measured TPA coefficient (β) of the device.

Reply:

We appreciate the constructive advice.

We supply the relevant data fitting in Fig. S9b of previous version with the equation:

$$I_{\text{phc}} = k_1 \cdot P^{\alpha_1} + k_2 \cdot P^{\alpha_2}, \quad (1)$$

where the first term on the right side represents the photocurrent from plasmonic resonance-induced hot-carrier, and the second term represents that from TPA photocarrier generation. The power exponents are calculated as $\alpha_1 = 0.86$ and $\alpha_2 = 1.94$, corresponding to linear and nonlinear processes respectively, as shown in Fig. R1. Note that $\alpha_1 < 1$ and $\alpha_2 < 2$, which can be attributed to the filling of trap-states with the increase of the input power.^{1,2}

For a small input power ($P < 100 \mu\text{W}$), $k_1 \cdot P^{\alpha_1} \gg k_2 \cdot P^{\alpha_2}$. Then, $I_{\text{phc}} \approx k_1 \cdot P^{0.86}$, suggesting that the plasmonic resonance-induced hot-carrier mechanism dominates the device performance. For a large input power ($P > 100 \mu\text{W}$), the nonlinear process is predominant, i.e., $I_{\text{phc}} \approx k_2 \cdot P^{1.94}$, proving that most photocurrent comes from the TPA photocarrier generation. Under the laser power of 2 mW, the contribution of the TPA photocarrier can be calculated by $(k_2 \cdot P^{\alpha_2})/I_{\text{phc}} \approx 90\%$.

Fig. R1 | Photocurrent of the device as a function of laser power at 1550 nm. The photocurrent can be fitted as $I_{\text{phc}} = k_1 \cdot P^{\alpha_1} + k_2 \cdot P^{\alpha_2}$, where $\alpha_1 = 0.86$, $\alpha_2 = 1.94$, $k_1 \approx 8.12 \times 10^{-7} \text{ A} \cdot \mu\text{W}^{-\alpha_1}$, $k_2 \approx 7.97 \times 10^{-10} \text{ A} \cdot \mu\text{W}^{-\alpha_2}$.

Traditionally, the TPA coefficient (β) can be measured by Z-scan,³ nonlinear transmittance,⁴ or two-photon induced fluorescence.⁵⁻⁷ For convenience, the last one is adopted, and the β can be obtained by measuring the fluorescence signal under single-photon excitation and two-photon excitation, without the involvement of many experimental parameters.

With an input laser beam of angular frequency ω , the intensity of the two-photon induced fluorescence can be written as

$$F_2 = \frac{1}{2} K \phi \beta \frac{P_2^2}{\tau f A \hbar \omega} L, \quad (2)$$

where ϕ is the fluorescence quantum efficiency, β is the TPA coefficient at the angular frequency of ω , P_2 is the average power of the input laser, τ is the pulse width, f is the repetition frequency, A is the effective area under the laser spot, L is the thickness of the sample, \hbar is the reduced Planck constant, and K is a dimensionless constant related to the system collection efficiency and self-absorption correction coefficient.

In the same optical system, with an input laser beam of angular frequency 2ω , the intensity of the single-photon induced fluorescence can be written as

$$F_1 = K\phi\alpha\frac{P_1}{2\hbar\omega}L, \quad (3)$$

where α is the single-photon absorption coefficient at the angular frequency of 2ω . Combining the Eqs. (2,3), the TPA coefficient can be written as

$$\beta = \frac{\tau f A F_2 P_1}{F_1 P_2^2} \alpha. \quad (4)$$

With a laser of $\tau = 2$ ps, $f = 76$ MHz and $A = 0.54 \mu\text{m}^2$ (with an objective of NA = 0.9), β can be obtained from Eq. (4) after PL intensities of two-photon and single-photon excitations and α are measured. For example, with certain excitation powers, the PL spectra of the MoS₂/WSe₂ heterostructure on SiO₂/Si under 1064 and 532 nm excitations are shown in Fig. R2a. For different laser powers, the integrated PL intensities are given in Fig. R2b. The α of the MoS₂/WSe₂ heterostructure at 532 nm is measured about $2.5 \times 10^5 \text{ cm}^{-1}$.

Fig. R2 | **a**, PL spectra of the MoS₂/WSe₂ heterostructure on SiO₂/Si with the excitation laser of 1064 nm (1 mW) and 532 nm (10 μW). **b**, PL intensity of the MoS₂/WSe₂ heterostructure under different laser powers with the excitation wavelengths of 1064 nm and 532 nm.

From Fig. R2b, the β is calculated as $6 \times 10^3 \text{ cm/GW}$ at 1064 nm when the input power is 2 mW, close to those reported before.^{8,9} The β at other wavelengths can be estimated (with $G_{\text{TPA}} = \frac{\beta}{2\hbar\omega A^2} \cdot P_{\omega}^2$) by comparing the TPA induced photocurrent (Fig. 2e in the manuscript) of the heterostructure on SiO₂/Si at different wavelength. Then, $\beta = 1130 \text{ cm/GW}$ at 1550 nm.

For the MoS₂/WSe₂ heterostructure on the metasurface, the TPA process is enhanced by the large field enhancement. The effective TPA coefficient, β_{eff} can be got after taking account of the carrier mobility, plasmonic resonance-induced hot-carrier, and exciton-based interface engineering. The enhancement of TPA photocarrier generation is evaluated as ~ 12500 -fold with the data from Paragraph 3 in the ‘‘Performance of the photoelectric enhancement’’ part of main text. Then, the $\beta_{\text{eff}} = \beta \times 12500 \approx 1.4 \times 10^7 \text{ cm/GW}$ at 1550 nm.

Revisions:

In the revised version, the fitting of the power-dependent photocurrent data and related discussions are supplied to the Section 13 of supplementary information (Section 9 of the previous version).

2. Photodetector figure of merit

The high responsivity is achieved under external bias, which invariably increases the dark current and noise. Performance metrics must account for this trade-off. The authors must perform noise measurements and a detectivity assessment. Please report the noise level (e.g., noise equivalent power, NEP) and the corresponding specific detectivity (D^*) and EQE.

The external bias introduces photoconductive gain. Please discuss the gain mechanism and report the measured responsivity bandwidth (speed) of the device, as high gain often trades off with speed.

Reply:

In order to calculate the noise equivalent power (NEP) of the photodetector, we measure the dark current of the device with a source-drain voltage $V_{ds} = 1$ V and back-gate voltage $V_g = -5$ V (the same as Fig. 2d,e). After fast Fourier transform, the noise power spectrum can be obtained, as shown in Fig. R3a. At 1 Hz, we get the noise current spectral density of the detector $S(f = 1\text{Hz}) = 1.21 \times 10^{-12}$ A/Hz^{1/2}. The specific detectivity D^* in this circumstance can be calculated as follows¹⁰⁻¹²

$$D^* = \frac{\sqrt{AB}}{\text{NEP}} = R \frac{\sqrt{A}}{S}, \quad (5)$$

where R is the responsivity, A the active area of the detector, and B the noise bandwidth. At $R = 1.35$ A/W at 1550 nm, the NEP is 8.96×10^{-13} W/Hz^{1/2}. With the measured R in the NIR-II window shown in Fig. 2e, the D^* at different wavelengths can be obtained, as shown in Fig. R3b. At 1550 nm, the D^* reaches 2.08×10^8 Jones. The external quantum efficiency (EQE) of the device can be estimated by the formula $\text{EQE} = hcR/e\lambda$, where h , c , e , and λ represent the Planck constant, the speed of light in the vacuum, elementary charge and wavelength, respectively. Figure R3c shows the EQE of the photodetector in the NIR-II window, reaching 108% at 1550 nm. The EQE that exceeds 100% is attributed to the large photoconductive gain of the device.

Fig. R3 | **a**, Noise power density of the dark current of the photodetector measured for $V_{ds} = 1$ V and $V_g = -5$ V, corresponding to the same conditions in which we measured its responsivity. **b**, Specific

detectivity (D^*) and **c**, EQE of the photodetector.

Figure R4 shows the V_{ds} -resolved and time-resolved photoresponse of photodetector under input laser of 1550 nm. The photocurrent shows a linear dependence on the bias voltage (Fig. R4a), suggesting higher responsivity can be readily achieved by applying a larger bias voltage, which indicates a larger photoconductive gain. The gain of the detector is defined as $G = \tau_{life}/\tau_{transit}$,¹² where τ_{life} is the lifetime of the carrier and $\tau_{transit}$ is the transiting time of the carrier. We first consider the $\tau_{transit}$, which is inversely proportional to the electron mobility and can be defined as $\tau_{transit} = L^2/\mu V_{ds} \sim 11$ ns, where L is length of the channel and μ is the mobility. From the temporal response of the photodetectors shown in Fig. R4b, the τ_{life} is estimated on the order of 300 ms. As a result, multiple electrons are recirculated in the heterostructure channel following a single electron-hole photo-generation, leading to a photoconductive gain on the order of 10^7 .

In such a regime where $\tau_{life} \gg \tau_{transit}$, the responsivity speed is primarily limited by τ_{life} . A shorter τ_{life} leads to faster response, whereas a longer τ_{life} enhances the photoconductive gain. This highlights a fundamental trade-off between the response speed and the gain in the photoconductive operation of the detector.

Fig. R4 | **a**, The magnitude of the photocurrent increases linearly with source–drain bias voltage under input laser (1550 nm) of different power. $V_g = -5$ V. **b**, Time-resolved photoresponse of the device, recorded for different values of bias voltage V_{ds} with input power (1550 nm) of 2 mW. $\tau_{rise} \approx \tau_{decay} \approx 300$ ms.

Revisions:

In the revised version, the noise measurement and detectivity assessment (including Fig. R3) are added as the Section 15 of supplementary information.

In the revised version, the discussion of the gain mechanism, together with the responsivity speed (including Fig. R4), are added as the Section 16 of supplementary information.

In the revised version, the sentences “Besides, the noise equivalent power.....the external quantum efficiency (EQE) reaches 108% at 1550 nm.” are added to the ending of the Paragraph 3 in the “Performance of the photoelectric enhancement” part of main text.

3. The claim that the photonic mode employed in this work is an “anapole state” is questionable.

The analysis or discussion about this claim is not solid. “Nonradiative” is not an exclusive characteristic of an anapole state. In fact, the photonic mode employed in this work seems like a typical light funneling mode (or a plasmonic cavity mode), which has been explicitly studied. Please check the following papers about this mode: PRL 107, 093902 (2011), Nat. Commun. 7, 11283 (2016). If the authors want to keep the claim about “anapole state”, please provide more substantial analysis.

Reply:

The anapole state is not typically classified as a fundamental eigenmode of a system. Instead, it usually represents a special, highly localized electromagnetic field distribution arising from the destructive interference between the toroidal dipole (TD) and electric dipole (ED) moments.^{13,14} This unique phenomenon, which leads to a suppression of far-field radiation, can be realized in both plasmonic cavities¹⁵ and dielectric nanostructures.¹⁶ Therefore, anapole state can also exist in our quasi-BIC mode.

The manipulation of light at the nanoscale heavily relies on the interference and interplay among multipolar families, primarily the electric, magnetic, and toroidal moments. In the far field zone, toroidal multipoles produce the same radiation patterns as corresponding electric or magnetic multipoles. This similarity allows for destructive interference between, for example, a TD and an ED. When their radiated fields are out-of-phase and of equal strength, they cancel each other in the far field, leading to a pronounced dip in the scattering spectrum. This phenomenon is known as the anapole state. Despite the vanishing far-field scattering, the electromagnetic fields remain strongly confined within the nanostructure itself. Therefore, the most definitive signature of the fundamental anapole state is the complete destructive interference between the TD and the ED moments. As shown in Fig. 1c, the destructive interference between TD and ED happens in our device, indicating the existence of anapole states. Next, we will carefully compare the differences between modes in the two papers you provided^{17,18} and that in our work.

For the light funneling mode reported in Ref. 17, it corresponds to our gap plasmon polariton (GPP) mode and regards to the understanding of the physical mechanism. The funneling in Ref. 17 is ascribed to the magnetoelectric interference of the incident wave with the evanescent field. However, above understanding of light funneling may stop at a GPP mode supported by a one-dimensional groove array, which is dominated by the ED component. While in our structure with more degrees of freedom, the case becomes complicated and the optical quasi-BIC mode originates from the coupled GPP modes in a 2D array.¹⁹ By multipole decomposition, the anapole state is found to drive the dominance of higher-order multipoles.

Although the plasmonic cavity mode in Ref. 18 also employs a 2D array, its enhancement mechanism is fundamentally different from ours. Multipole decomposition of the structure in Ref. 18 is carried out, as shown in Fig. R5a. The resonance is obviously dominated by ED, which is much larger than TD. Therefore, only an incomplete destructive interference happens between TD and ED. While the destructive interference between TD and ED in our structure is more complete, and the ED+TD is much nearer to zero, as shown in Fig. 1c. Focusing on the higher-order multipoles, the multipole decomposition at logarithmic coordinate of the structure in Ref. 18 is given in Fig. R5b. Comparing the radiative

components, it can be seen that the ED+TD component is more than one order of magnitude larger than EQ component, and more than two order of magnitude larger than EO component, i.e. $ED+TD \gg EQ \gg EO$. While our structure shows significant differences, as shown in Fig. 1c (To discriminate more clearly the different multipoles, the multipole decomposition of the optical mode in our structure is displayed at logarithmic coordinate in Fig. 1c). $EQ > EO > ED+TD$ here, powerfully proving the anapole-driven dominance of higher-order multipoles in our structure.

Fig. R5 | **a**, Scattering cross section of the structure in Ref. 18, which is simulated by the parameters obtained from Ref. 18. The gray line labels the resonance wavelength of the optical mode. **b**, Scattering cross section at logarithmic coordinate of the structure in Ref. 18.

In a word, the criterion for establishing an anapole state is the occurrence of complete destructive interference between the ED and TD within the optical mode. Figure R6 shows the scattering cross section of ED, TD and ED+TD and the phase difference between the ED and TD. It can be found that, at the resonance wavelengths, the ED and TD moments have inverse phases and the same magnitude, thus complete destructive interference will happen and result in the anapole state. In practice, since the phase difference may not be exactly π and the ED+TD scattering does not vanish entirely, we define the anapole state as existing when the condition $ED+TD \ll ED, TD$ is satisfied. Therefore, the existence of anapole state requires creating certain conditions to implement and achieve a goal of making high-order multipoles dominant.

Fig. R6 | Scattering cross section of ED, TD and ED+TD and the phase difference between the ED and TD. At the resonance wavelengths (cyan and orange areas), the ED and TD cancel each other the most

and a minimal ED+TD is achieved.

Revisions:

In the revised version, Fig. 1c is recalculated and redrawn in the logarithmic coordinate system.

In the revised version, Fig. R6 and the related discussion are added as the Section 6 of supplementary information.

In the revised version, the words “(see Fig. S6 for details)” are added to the Line 21, Paragraph 1 in the “Basic structure and properties of the nanostructured device” part of main text.

4. Relationship between “quasi-BIC” and “anapole state”

Q1: The physical mechanism explanation must be significantly strengthened by clarifying the interplay between the quasi-BIC mode and the anapole state. Please explicitly define the causal relationship (or parallel relationship) between the quasi-BIC and the anapole state. Is the quasi-BIC a necessary consequence of the anapole-driven dominance of higher-order multipoles?

Q2: The authors should clearly describe how the perfect BIC mode is identified before symmetry breaking (Δy). A supplementary analysis showing the asymmetry parameter (Δy) dependent Q-factor distribution and/or the far-field polarization singularity would be highly beneficial in validating the claimed perfect BIC and the evolution of quasi-BIC.

Q3: What key benefits does the anapole state offer over other high-Q photonic modes (e.g., guided mode resonance, chiral BIC) in the context of boosting photodetection efficiency?

Reply:

We are grateful for the valuable comments, and will answer the three questions separately.

To Q1: There is a parallel relationship between the quasi-BIC and the anapole state, which work together to achieve a large field enhancement. The quasi-BIC is not a necessary consequence of the anapole state, but help promoting higher-order multipoles.

For the symmetric structure ($\Delta y = 0$ nm), a unit cell of which is shown in Fig. R7a. Such a periodic metasurface supports GPP mode under x -polarized input light (the resonance shown in Fig. S4 at $\Delta y = 0$ nm). The blue and orange small spheres represent concentrated negative and positive charges within the structure, which is consistent with the z -component of the electric field shown in Fig. R7b. The blue arrows indicate the electric field lines pointing from the positive charges to the negative ones, which constitute the ED moments. Similarly, the green arrows stand for magnetic dipole (MD) moments. Under certain conditions, magnetic loops can take shape, forming TD moments at the center of the loop,²⁰ shown by the red arrows. A destructive interference between ED and TD moments results in the anapole state, which is verified by the multipole decomposition of the GPP mode as shown in Fig. R7c.

Fig. R7 | Construction of high-order multipoles in metasurfaces. **a**, Illustration of different dipole moments of GPP mode in symmetric structure ($\Delta y = 0$ nm) within one unit cell. The green, blue, and red arrows stand for MD, ED, and TD moments, respectively. **b**, Simulated z -component of electric and magnetic fields on the $z = 0$ surface for GPP mode. **c**, The scattering cross-section of different multipoles of GPP mode. At the resonance wavelength, the ED and TD under the Cartesian coordinate system cancel each other the most and a minimal ED+TD is achieved. **d**, Illustration of different dipole moments of quasi-BICs in asymmetric structure ($\Delta y \neq 0$ nm) within one unit cell. **e**, Simulated z -component of electric and magnetic fields on the $z = 0$ surface for quasi-BICs. **f**, The scattering cross-section of different multipoles of quasi-BIC mode.

It is obvious that the anapole state has existed and EO has exceeded ED+TD in GPP mode. However, the components of high-order multipoles are still low. By breaking the symmetry of the structure, quasi-BICs take place,²¹ as shown in Fig. R7d,e. The charge distributions become uneven with asymmetric characteristics. The BICs with zero net ED moment cannot be excited. Whereas, the quasi-BICs can be excited with the aid of symmetry breaking. As shown in Fig. R7f, the EQ component dominates the quasi-BIC mode, i.e. $EQ \gg ED+TD$. In Fig. R7c, for the GPP mode, the charges at the four corners in the top and bottom planes display net ED moments respectively but with opposite signs, and only the EQ_{xz} constituent exists. While for the quasi-BIC mode in Fig. R7f, the charges on the four corners in the top and bottom planes show obvious EQ features, respectively, and a larger EQ_{xy} constituent can be found.

In summary, from the perspective of multipole modulation, the anapole state suppresses the ED component, while the quasi-BIC enhances the EQ component. Quantitatively, the anapole state reduces the ED by three orders of magnitude (comparing ED and ED+TD in Fig. R7c), and the quasi-BIC increases the EQ by two orders of magnitude (comparing EQ in Fig. R7c and that in Fig. R7f). The suppression of the fundamental ED by the anapole state is a prerequisite for the dominance of higher-order multipoles. As evidenced in Fig. R7f, without the destructive interference that forms the anapole

state, the ED remains stronger than the EQ even in the presence of the quasi-BIC mode.

To Q2: Thank you for the helpful suggestion. The supplementary simulation is given in Fig. R8.

Fig. R8 | Evidence of perfect BIC. **a**, Δy dependent Q-factor. **b**, Asymmetry parameter dependent far-field polarization. To make Δx meaningful, the P is enlarged to 1200 nm in this simulation.

As shown in Fig. R8a, Q factor of optical mode in our structure increases with the decrease of Δy , and tends to $+\infty$ at $\Delta y = 0$, demonstrating a typical BIC feature at the high-symmetry point.²¹ The far-field eigenpolarization map of the resonant mode is shown in Fig. R8b, and a singularity appears at the high-symmetry point, which is another typical feature of the perfect BIC.²²

To Q3: Among the nanostructure, the field enhancement can be written as $|E_\omega/E_0|^2 \propto \gamma_{\text{rad}}/[V(\gamma_{\text{rad}} + \gamma_{\text{diss}})^2]$, where E_ω and E_0 denote the enhanced electric field and the input electric field, V is the effective mode volume, and γ_{rad} and γ_{diss} represent the radiative and dissipative losses of the system. The Q factor is inversely proportional to the mode losses. For the quasi-BIC, the Δy regulates the radiative loss γ_{rad} . For a certain dissipative loss γ_{diss} , which is highly dependent on the inherent loss of materials, $\gamma_{\text{rad}}/(\gamma_{\text{rad}} + \gamma_{\text{diss}})^2$ could reach the maximum at the $\gamma_{\text{rad}} = \gamma_{\text{diss}}$ condition. As for the anapole states, the anapole-driven dominance of higher-order multipoles brings extreme small V in the structure, which is more beneficial to the field enhancement.

Quantitatively, we compare the following several situations: without anapole & be not quasi-BIC (taking Ref. 18 as an example, Fig. R5); with anapole & be not quasi-BIC (the GPP mode at $\Delta y = 0$ nm of our structure, Fig. R7a–c); with anapole & be quasi-BIC (the quasi-BIC mode at $\Delta y = 100$ nm of our structure, Fig. R7d–f). The simulations are carried out with the same mesh size.

Comparing the electric field distribution of the structures without and with anapole state, as shown in Fig. R9a,b, the maximum electric field of the optical mode with anapole state is more than 3-fold larger than that without anapole state. And the field in Fig. R9b is much more localized than that in Fig. R9a. That is because that the optical mode in Fig. R9a is dominated by ED component, as shown in Fig. R5b. While the optical mode in Fig. R9b is dominated by higher-order multipoles, as shown in Fig. R7c, which is attributed to the anapole state. Then, comparing the electric field distribution of the structures of GPP and quasi-BIC mode, the electric field increases for about 50%, which comes from the loss modulation and the promoting of EQ (Fig. R7c,f).

Fig. R9 | Comparing of electric field enhancement. **a**, Calculated electric field distribution of one unit of the structure in Ref. 18. **b**, Calculated electric field distribution of one unit of our structure at $\Delta y = 0$ nm. **c**, Calculated electric field distribution of one unit of our structure at $\Delta y = 100$ nm.

In summary, the smaller V brought by high-order multipoles (induced by anapole state) is the core strength of anapole state over the others. More in-depth theoretical analysis of the relation between high-order multipoles and V can be referred to in our previous works.^{19,23}

Revisions:

In the revised version, a sentence “The detailed analysis and comparisons of the optical modes can be referred to in our previous works.^{46,47}” is added to the ending of the Paragraph 1 in the “Basic structure and properties of the nanostructured device” part of main text.

In the revised version, Fig. R8 and the related discussion are added as the Section 5 of supplementary information.

In the revised version, the words “(Fig. S5)” are added to the Line 8, Paragraph 1 in the “Basic structure and properties of the nanostructured device” part of main text.

5. Figure clarity

Figure 1a contains too many sub-panels. The authors should re-organize or re-design this figure to ensure that all labels are clear, legible, and that accurate, unambiguous citation of individual panels in the main text is feasible.

Reply:

Thank you for the useful suggestion.

The labels in Fig. 1a are re-designed and are clearer now.

Revisions:

In the revised version, Fig. 1a is re-designed and redrawn. The sub-panels are renamed as a.1, a.2, a.3, a.4, and the related descriptions in the main text are revised, respectively.

6. Fabrication and device structure description

Clarification on the device architecture, particularly regarding the insulation layers, is needed to validate the operational mechanisms. In Supplementary Information, Section 1, the text mentions: “To ensure the pad film continuity between top of single-crystalline Ag and SiO₂/Si substrate, the edges of the single crystalline Ag were cut by FIB and covered by a thick SiO₂ film of 200 nm. The other SiO₂ film with thickness of 100 nm positioned between Au substrate and Au electrodes acted as an insulation layer.” Please clarify the position and function of the “Au substrate.” Is the blue layer in the schematics (e.g., Fig. 1a) the SiO₂ film? How was this 100 nm SiO₂ film fabricated?

Could the authors please clarify the exact location of the “thick SiO₂ film of 200 nm” within the device structure? The authors state this SiO₂ only separates the Au contact and Ag metasurface. If the MoS₂/WSe₂ heterostructure is also separated from the metasurface (Ag) by this SiO₂ layer, the contribution of hot-carrier injection from Ag would be fully excluded. Could the authors confirm the presence of any silver oxide layer (e.g., Ag₂O) on the surface of the single-crystalline Ag metasurface?

Reply:

Thank you for the significant comment.

We have made a mistake here. The “Au substrate” should be “Ag substrate” in the sentence “The other SiO₂ film with thickness of 100 nm positioned between Au substrate and Au electrodes acted as an insulation layer.” And it is corrected in the revised version.

The blue layer in the schematics of Fig. 1a is the SiO₂ film, which was fabricated by magnetron sputtering.

Figure S1c is redrawn as shown in Fig. R10, where the location of “thick SiO₂ film of 200 nm” is labeled. Different from the SiO₂ insulation layer between Au electrode and Ag substrate, the SiO₂ step of 200 nm is used to bridge the height difference between Ag plate and the SiO₂/Si substrate.

Fig. R10 | Redrawing of Fig. S1c.

The single-crystalline Ag is synthesized by polyol reduction method in liquid environment, and is coated by polyvinyl pyrrolidone (PVP) during and after synthesis, which plays a role of preventing oxidation or sulfuration. Following the fabrication of the target structures, the covering layer is broken. The structures are then immediately encapsulated by transferred two-dimensional materials, which effectively isolates the Ag metasurface from ambient air. Furthermore, we have confirmed the

spotlessness of the Ag metasurface by successfully forming a self-assembled monolayer (SAM) of molecules on the surface of Ag via chemical bond. The 4-aminothiophenol (4-ATP) was used in this experiment. The -SH bond in 4-ATP can react with Ag and form Ag-S bond. By immersing the substrate in the 4-ATP solution of 10^{-4} M for ~ 10 h, a layer of continuous, even, dense SAM can take shape on the surface of Ag structure,²⁴ as illustrated in Fig. R11a. Afterwards, the sample was rinsed several times with ethanol and deionized water to remove excessive molecules and dried up with cleaned nitrogen. Then, the Raman scattering spectra were measured on the metasurface. The strong Raman signal can be observed, as shown in Fig. R11b, verifying the formation of SAM, which demonstrates a clean Ag surface.

Fig. R11 | **a**, SAMs on Ag surface. **b**, Raman spectrum of 4-ATP molecule on Ag metasurfaces.

Revisions:

In the revised version, the sentence “The other SiO₂ film with thickness of 100 nm positioned between Au substrate and Au electrodes acted as an insulation layer.” is replaced by “The other SiO₂ film with thickness of 100 nm positioned between **Ag** substrate and Au electrodes acted as an insulation layer.” in the **Section 1** of **supplementary information**.

In the revised version, **Figure S1c** is redrawn, as shown in Fig. R10, where the locations of “**SiO₂ step of 200 nm**” and “**SiO₂ insulation layer of 100 nm**” are labeled.

In the revised version, a paragraph “**For the stability of the single-crystalline Ag.....over one month without performance degradation.**” is added to the **Section 1** of **supplementary information**.

7. References

There are many previous works about using metallic nanostructures to enhance hot-electron photoresponse. Some recent works include InfoMat.6, e12556 (2024), Nat. Electron. 7, 1004 (2024), Light Sci. Appl. 12, 176 (2023)·· Please pay attention to the previous works in this field and make some discussion.

Reply:

We appreciate the meaningful recommendation.

We acknowledge the seminal contributions of prior works on hot-carrier injection (e.g., Refs. 25–27), which have laid a critical foundation for this field and paved the way for various applications, spanning high-performance photodetection, sub-bandgap response, solar energy harvesting, and multidimensional detection.

In our work, the role of the giant field enhancement is twofold: TPA process, and plasmonic resonance-induced hot carriers. As quantified in our response to Comment 1, approximately 90% of the photocurrent at 1550 nm is attributed to TPA-generated carriers, with the remaining 10% originating from the plasmonic resonance-induced hot-carrier injection. Therefore, while hot carriers constitute a secondary contribution, they remain a valuable and complementary part of the overall photoresponse mechanism, particularly under low-power excitation.

We have revised the manuscript to include a discussion of these prior works and to clearly delineate the distinct mechanisms at play. We believe this revision better contextualizes our work and more effectively highlights its novelty and significance. Thank you again for this constructive suggestion, which has greatly strengthened our paper.

Revisions:

In the revised version, the sentences “**Independent of the semiconductor's bandgap, hot carrier injection.....selectively discriminating optical signals based on chirality, phase, and polarization.**^{57,58}” are added to the ending of the “Performance of the photoelectric enhancement” part of **main text**.

In the revised version, the Refs. 25–27 in the response letter are added to the main text as **Refs. 56–58**.

8. Typo

In Equation (2), the term “ $I_{\text{phc msx}}$ ” should be corrected to “ $I_{\text{phc max}}$ ”.

Reply:

Thank you for the reminder. The “ $I_{\text{phc msx}}$ ” is corrected to “ $I_{\text{phc max}}$ ” in the revised version.

Revisions:

In the revised version, the “ $I_{\text{phc msx}}$ ” is corrected to “ $I_{\text{phc max}}$ ” in **Eq. (2)**.

To Reviewer #2:**Comments:**

Reply:

We sincerely appreciate your time and effort in reviewing our manuscript.

To Reviewer #3:

Comments:

In this manuscript, the authors successfully leverage quasi-BIC and higher-order multipoles to achieve exceptional electric field enhancement exceeding one thousand times. This work represents significant progress toward miniaturized multifunctional photodetectors for biological imaging and communication applications. However, several key issues as follow require clarification before the manuscript can be fully endorsed for publication.

Reply:

We are grateful for your positive assessment of our work. The valuable suggestions have significantly improved the manuscript and will benefit future research. We have carefully addressed all the issues and made the necessary revisions in the updated manuscript.

(1) The authors claim that the field enhancement trends for SHG and TPA are identical due to their mutual dependence on the fourth power of the electric field. However, SHG represents a coherent process dependent on phase matching, while TPA constitutes an incoherent absorption process. Additionally, given their distinct nonlinear susceptibilities, it remains unclear whether assuming identical enhancement trends is fully justified, particularly in a system containing multiple excitonic and plasmonic resonances?

Reply:

Thank you for raising this insightful and important point regarding the fundamental differences between the coherent SHG and incoherent TPA processes, which prompts us to improve the strictness of this issue.

We completely agree that their distinct physical nature, particularly the phase-matching dependence of SHG, can, in general, lead to divergent enhancement trends. However, compared to bulk or waveguide cases, the phase-matching requirement is greatly relaxed for SHG in our atomically thin materials. The conversion efficiency here is predominantly dictated by the field enhancement, rather than by phase matching over a propagation length.

Therefore, while the nonlinear susceptibilities are indeed distinct, both processes share a common driving force: the local intensity at the pump frequency, which scales with $|E_\omega|^4$ in our model. Generally, the SHG power can be written as^{23,28}

$$P_{\text{SHG}} = \left(\chi_{\text{eff}}^{(2)}\right)^2 \cdot \frac{(\omega L)^2}{A \epsilon_0 c^3} \cdot P_\omega^2, \quad (6)$$

where P_{SHG} represents the power of SHG, ω the angular frequency of pump laser, L the length of the nonlinear material, A the laser area, ϵ_0 the vacuum permittivity, c the speed of light in vacuum, P_ω the power of the pump laser, and $\chi_{\text{eff}}^{(2)}$ the effective second-order nonlinear coefficient at the angular

frequency of ω . The above equation is derived based on some assumptions, especially, the phase-matching condition. Comparing the same materials on flat Ag and on Ag metasurface under the same optical system, the only variable is P_ω . With $P_\omega \propto |E_\omega|^2$, the enhancement of P_{SHG} can be written as $\text{EF}_{\text{SHG}} = P_{\text{en}}^2/P_0^2 = |E_{\text{en}}/E_0|^4$, where the subscript _{en} represents the enhanced parameter, and subscript ₀ that before enhancement. Note that, without resonance at the emitting wavelength, the emitting enhancements of materials on flat Ag and on Ag metasurface are seen as the same.

On the other hand, TPA carrier production rate can be written as²⁹

$$G_{\text{TPA}} = \frac{\beta}{2\hbar\omega A^2} \cdot P_\omega^2, \quad (7)$$

where G_{TPA} represents the TPA carrier production rate, β the TPA coefficient, $\hbar\omega$ the energy of a photon. Similar to Eq. (6), the enhancement of G_{TPA} can be written as $\text{EF}_{\text{TPA}} = P_{\text{en}}^2/P_0^2 = |E_{\text{en}}/E_0|^4$.

Then, it can be obtained that $\text{EF}_{\text{SHG}} = \text{EF}_{\text{TPA}} = |E_{\text{en}}/E_0|^4$.

Revisions:

In the revised version, the discussions about the consistency of the enhancement of SHG and TPA are added as the Section 11 of supplementary information.

In the revised version, the words “(see Section 11 in supplementary information for details)” are added to the ending of the Paragraph 1 in the “Performance of the photoelectric enhancement” part of main text.

(2) The approximately 58-fold photoresponse enhancement on flat silver is explained as the product of increased carrier mobility, field enhancement for two-photon absorption, and an exciton-recycling factor. The theoretical basis for treating these contributions spanning both linear and nonlinear processes as multiplicative requires further justification. Additional modeling or controlled experiments would help decouple these mechanisms and validate the proposed model.

Reply:

We are grateful for the insightful comment.

For the photoconductive photodetector in our manuscript, the photocurrent can be written as³⁰

$$I_{\text{phc}} = eG\eta_c\mu\tau A \frac{V_{\text{bias}}}{L} = e(G_{\text{TPA}} + G_{\text{hot}}) \cdot \eta_c\mu\tau A \frac{V_{\text{bias}}}{L}, \quad (8)$$

where e is the elementary charge, G the photogenerated carrier production rate, η_c the carrier collection efficiency, μ the carrier mobility, τ the carrier lifetime, V_{bias} the bias voltage, A the cross section of the device, L the channel length, G_{TPA} the TPA-induced carrier production rate, and G_{hot} the plasmonic resonance-induced hot-carrier production rate.

For the heterostructure on flat Ag or on SiO₂/Si substrates, with $G_{\text{TPA}} \propto |E_\omega|^4$ and $G_{\text{hot}} = 0$,

$$I_{\text{phc}} \propto |E_\omega|^4 \eta_c \mu. \quad (9)$$

Now consider the three variables on the right side of Eq. (9) one by one. First, the $|E_\omega|^4$ on the surface of perfect metal has an enhancement factor of ~ 16 comparing to that on SiO_2/Si , which can be verified experimentally. The difference of field enhancement can be compared measuring the SHG signal of the same material on these two substrates. The SHG signal is proportional to the $|E_\omega|^4 \cdot |E_{2\omega}|^2$,³¹ where $E_{2\omega}$ is the electric field at the SHG wavelength (emitting wavelength). For the flat Ag or SiO_2/Si without structure, the field enhancement at these two angular frequencies (ω , 2ω) can be seen invariable. Experimentally, the SHG signal of TMDC on flat Ag is about 50~80-fold larger than that on SiO_2/Si .

Second, the carrier mobility μ can be calculated by $\mu = [dI_{ds}/dV_g] \times [L/WC_i V_{ds}]$,³² where L , W and C_i are the length, width and capacitance of device channel, respectively. Experimentally, μ is calculated to be $\sim 87 \text{ cm}^2\text{V}^{-1}\text{s}^{-1}$ on single-crystal Ag and $\sim 43 \text{ cm}^2\text{V}^{-1}\text{s}^{-1}$ on SiO_2/Si .

Third, for the carrier collection efficiency η_c , it can be seen as the utilization efficiency of the carriers or excitons excited by the input laser. And the η_c on the metal surface is larger than that on SiO_2/Si , the difference of which can be calculated to be ~ 1.78 -fold. As shown in Fig. R12, the excitons excited on the Ag surface have many decaying channels, including direct separation, SPP excitation, electron-hole excitation, and photon emission, among which the energy decaying into Ag can reinject to 2D semiconductor as hot-carriers. While for the excitons excited on the SiO_2/Si surface, there are only the channels of direct separation and photon emission. Moreover, the photon emission part on SiO_2/Si is larger than that on Ag surface without the quenching effect. Therefore, the utilization efficiency of the excitons on Ag surface is larger than that on SiO_2/Si surface.

Fig. R12 | Exciton decaying channels at the interface of TMDCs/metal.³³

(3) While the authors note using a 2 ps pulsed laser for nonlinear measurements, the repetition rate remains unspecified. Providing the repetition rate and estimating the peak power density at the sample would enable accurate quantification of nonlinear efficiencies and facilitate comparisons with other studies.

Reply:

We appreciate the constructive suggestion.

The repetition of the pulsed laser is 76 MHz, as introduced in the “Methods” part of the previous manuscript. The peak power density can be calculated corresponding to input power. Taking the 1550 nm laser as an example, 1 mW of laser power corresponds to $\sim 0.7 \text{ GW}/\text{cm}^2$ of peak power density in

our manuscript.

Revisions:

In the revised version, the words “(repetition rate = 76 MHz)” are added to the Line 2, Paragraph 2 in the “Performance of the photoelectric enhancement” part of **main text**.

(4) Given the exceptionally high reported enhancement factors, information concerning the corresponding signal-to-noise ratios and device stability under continuous or pulsed operation would be valuable. The authors should also comment on whether any saturation or degradation effects were observed at operational power levels?

Reply:

Thank you for the useful recommendation.

For the noise measurements, the noise level and the corresponding specific detectivity can be referred to in the reply to Comment 2 from Reviewer #1. The noise equivalent power, NEP is $8.96 \times 10^{-13} \text{ W/Hz}^{1/2}$ in our device.

The device stability was tested under continuous illumination with a 1550 nm pulsed laser of $\sim 1 \text{ mW}$. As illustrated in Fig. R13a, time-resolved photoresponse measurements were performed after 1 h, 10 h, and 12 h of cumulative illumination. The device remained stable for the first hour. However, with prolonged exposure, its performance exhibited a decay before reaching a steady state after about 10 hours. The performance decay mainly comes from the large field enhancement at the resonance wavelength, which generates a large amount of heat under continuous illumination of 1 mW. Under a smaller input power, it may remain stable for a longer time.

Fig. R13 | **a**, Stability testing of the device. The device is continuously illuminated by the pulsed laser of 1550 nm for 1 h, 10 h and 12 h between the time-resolved photoresponse measurements, with the laser power of $\sim 1 \text{ mW}$. **b**, Saturation of the photoresponse under increased power of input laser.

The saturation behavior of the photodetector was investigated by measuring its performance under

increasing input laser power. As shown in Fig. S9a (previous version) and Fig. R1, the measured data begins to deviate from the fitting curve at high input powers. For clarity, the data from Fig. R1 is replotted on a linear scale in Fig. R13b. A clear discrepancy is observed for input powers exceeding 2 mW, where the measured photocurrent is significantly lower than the fitted values, indicating the onset of device saturation.

Revisions:

In the revised version, the noise measurement and detectivity assessment (including Fig. R3) are added as the Section 15 of supplementary information.

In the revised version, a sentence “Besides, the measured data begins.....the onset of device saturation.” is added to the ending of Section 13 of supplementary information.

(5) Further elaboration on the energy alignment between plasmon-generated hot carriers and the band structure of the heterostructure would enhance understanding of the underlying mechanisms. Specifically, how the Schottky barrier at the WSe₂ and silver interface influences hot carrier injection efficiency under different gate voltages warrants detailed discussion?

Reply:

We are thankful for the constructive guidance.

To investigate the gate-tunable Schottky barrier at the Ag/WSe₂ interface, we decomposed the photocurrent by leveraging the distinct power dependencies of the two dominant generation mechanisms: hot-carrier injection (linear) and TPA (quadratic). As detailed in our response to Comment 1 from Reviewer #1, the photocurrent of our device can be written as Eq. (1): $I_{\text{phe}} = k_1 \cdot P^{\alpha_1} + k_2 \cdot P^{\alpha_2}$, where the exponent $\alpha_1 \sim 1$ corresponds to the linear hot-carrier process, and $\alpha_2 \sim 2$ corresponds to the nonlinear TPA process.

We measured the power-dependent photocurrent under different gate voltages (V_g), as shown in Fig. R14a. By fitting the data to Eq. (1), we extracted the coefficients k_1 and k_2 , which quantitatively represent the efficiency of the hot-carrier injection and TPA contributions, respectively (Fig. R14b). The strong modulation of k_1 with V_g directly evidences the gate control over the Schottky barrier height (Φ_B). At $V_g = 0$ V, the small k_1 indicates a high Φ_B , which strongly suppresses hot-hole injection. When a negative gate voltage ($V_g = -5$ V) is applied, k_1 increases substantially, signifying a lowering of Φ_B that efficiently promotes the injection of hot holes from the Ag metasurface into WSe₂. This delineation not only confirms the gate-tunable nature of the interface barrier but also explicitly demonstrates how its modulation dictates the hot-carrier contribution to the overall photoresponse, solidifying our understanding of the device's operational principle.

Fig. R14 | **a**, Photocurrent of the device as a function of laser power at 1550 nm under different V_g . The photocurrent can be fitted as $I_{\text{phc}} = k_1 \cdot P^{\alpha_1} + k_2 \cdot P^{\alpha_2}$, where $\alpha_1 \sim 1$, $\alpha_2 \sim 2$. **b**, Fitted k_1 and k_2 under different V_g . The increased k_1 with a larger negative V_g indicates the decreased Schottky barrier under negative V_g .

Revisions:

In the revised version, the modulation of Schottky barrier by gate voltage and relative discussions (including Fig. R14) are added as the Section 17 of supplementary information.

(6) Although the quasi-BIC provides a high quality factor and strong field confinement, its narrowband nature raises questions about its impact on broadband photodetection performance. The authors should address whether there exists a design trade-off between resonance sharpness and operational bandwidth?

Reply:

Thank you for the valuable comment.

The primary objective in modulating the quasi-BIC is not to pursue an absolute maximum Q-factor, but to achieve the optimal condition where the radiative and dissipative loss rates are balanced ($\gamma_{\text{rad}} = \gamma_{\text{diss}}$, as detailed in our response to Comment 4. Q3 from Reviewer #1). Different from the all-dielectric metasurface, whose performance comes from high Q factor, the large field enhancement of the plasmonic metasurface is mainly depended on the small mode volume. At the optimized point, both the quasi-BIC and the coupled GPP mode exhibit a linewidth of approximately 150 nm (Fig. 3a), which is already narrower than that of the normal GPP mode (Fig. 3b). Consequently, the quasi-BIC provides field enhancement over the 1350–1650 nm range, while the GPP mode covers the 1050–1350 nm range (Fig. 2c).

In a word, the modulation of the quasi-BIC is to satisfy the $\gamma_{\text{rad}} = \gamma_{\text{diss}}$ condition, and the broadband photoresponse is naturally enabled via the synergistic effect of the two resonant modes.

Revisions:

In the revised version, the sentences “Unlike all-dielectric metasurfaces.....exhibit broadband enhancement with relatively low Q factor.” are added to the Line 13, Paragraph 2 in the “Performance of the photoelectric enhancement” part of main text.

(7) The critical importance of using single-crystalline silver to the observed performance requires further examination. Have simulations or experimental comparisons been conducted with polycrystalline silver or other plasmonic materials to evaluate the role of material loss?

Reply:

The polycrystalline Ag film evaporated by magnetron sputtering is used to compare with the single-crystalline Ag here.

It is well known that the metallic loss greatly influences field enhancement performance. We adopt single-crystalline Ag grown in liquid environment, which has much smoother surface than coated polycrystalline Ag film, as shown in Fig. R15a,b. The RMS roughness of them are about 0.3 and 2.7 nm, respectively. On the one hand, it is difficult to fabricate an eligible nanostructure on coated Ag film. Thus, the material loss of single-crystalline Ag is less than that of coated Ag film.

Fig. R15 | **a**, AFM image of polycrystalline Ag film synthesized by magnetron sputtering. **b**, AFM image of single-crystalline Ag synthesized by polyol reduction method in liquid environment. Scale bar, 500 nm. **c**, Reflectance of metasurface etched on polycrystalline and single-crystalline Ag. Inset: SEM images of the structures. **d**, Normalized spectra of fundamental laser (right, 1550 nm) and SHG signal (left, 775

nm) generated from the MoS₂/WSe₂ heterostructure on polycrystalline and single-crystalline Ag metasurface.

Quantitatively, the performances of the same structure (the same as that in Fig. 2) fabricated on the single-crystalline and polycrystalline Ag are compared, as shown in Fig. R15c,d. On the one hand, the Q factor of the single-crystalline Ag metasurface is larger than the other (Fig. R15c), indicating a smaller material loss of the single-crystalline Ag. On the other hand, the SHG enhancement of the single-crystalline Ag metasurface is about 8-fold larger than that of the polycrystalline Ag metasurface (Fig. R15d), proving a larger field enhancement of the single-crystalline Ag metasurface.

Therefore, single-crystalline Ag is of great significance to the device's performance.

Revisions:

In the revised version, the comparisons between polycrystalline and single-crystalline Ag and relative discussions (including Fig. R15d) are added as the Section 10 of supplementary information.

In the revised version, the words “(Section 10 in supplementary information)” are added to the Line 16, Paragraph 2 in the “Performance of the photoelectric enhancement” part of main text.

To Reviewer #4:

Comments:

The authors reported a surface-enhanced 2D TMDC photodetector by means of high-order multipoles with anapole states, as well as quasi-bound states in the continuum, to operate efficiently in the near-infrared second (NIR-II) window at room temperature. Both high-sensitive and chiral discrimination are achieved. The experiments and results are interesting. However, there are some concerns that should be further clarified.

Reply:

We are grateful for your valuable time and careful consideration of our manuscript. We have meticulously addressed all comments and revised the paper accordingly.

1. An objective and comprehensive detection performance comparison is necessary as the authors have emphasized the advantages of the performance of their devices and it is also important to verify the advantages of anapole enhancement, especially comparing with other plasmonic structures.

Reply:

Thank you for the reviewer's valuable feedback.

As detailed in our response to Comment 3 from Reviewer #1, the anapole-inspired metasurface in our manuscript is compared to the two similar structures in Refs. 17,18. And the simulated field enhancements of the structure in Ref. 18 and ours are compared in Fig. R9. To be more persuasive, we compare a GPP mode at the same mode wavelength with the quasi-BIC mode in our manuscript to verify the advantages of our structure, as shown in Fig. R16.

Fig. R16 | a, Reflectance of metasurface of GPP mode and quasi-BIC mode. Inset: SEM images of the

structures. **b**, Normalized spectra of fundamental laser (right, 1550 nm) and SHG signal (left, 775 nm) generated from the MoS₂/WSe₂ heterostructure on single-crystalline Ag metasurface of GPP mode and quasi-BIC mode.

The quasi-BIC mode exhibits a higher Q factor than the GPP mode (Fig. R16a), a characteristic mediated by its distinct loss modulation. Since the device performance relies on field enhancement, we compare the SHG enhancement for both modes. As shown in Fig. R16b, the quasi-BIC mode yields an SHG enhancement approximately 40 times greater than that of the GPP mode, confirming a stronger field confinement in our structure. The specific role of the anapole states in contributing to this enhancement is discussed in detail in our response to Comment 4 from Reviewer #1. The advances of our structure comparing with other structures are listed in detail in our previous work.^{17,21}

Revisions:

In the revised version, a sentence “The detailed analysis and comparisons of the optical modes can be referred to in our previous works.^{46,47}” is added to the ending of the Paragraph 1 in the “Basic structure and properties of the nanostructured device” part of main text.

2. Why the metal Ag was selected to fabricate the nanostructure? The metallic silver seems to be not very stable and is prone to oxidation. Will this affect the performance of the device?

Reply:

Thank you for the insightful comment.

Firstly, the metal Ag has a lower intrinsic loss comparing to other metals like Au, which is more beneficial to the field enhancement. The field enhancement of nanostructures can be written as $|E_\omega/E_0|^2 \propto \gamma_{\text{rad}}/[V(\gamma_{\text{rad}} + \gamma_{\text{diss}})^2]$, where E_ω and E_0 denote the enhanced electric field and the input electric field, V is the effective mode volume, and γ_{rad} and γ_{diss} represent the radiative and dissipative losses of the system. For a certain dissipative loss γ_{diss} , $\gamma_{\text{rad}}/(\gamma_{\text{rad}} + \gamma_{\text{diss}})^2$ could reaches the maximum at the $\gamma_{\text{rad}} = \gamma_{\text{diss}}$ condition, where $|E_\omega/E_0|^2 \propto 1/4\gamma_{\text{diss}}$. Then, the lower γ_{diss} in the metal Ag brings a larger field enhancement. The performance of structures based on polycrystalline and single-crystalline Ag are compared in Fig. R15, proving the advantages of the single-crystalline Ag.

The single-crystalline Ag is coated by polyvinyl pyrrolidone (PVP) after synthesis, offering the Ag superior oxidation or sulfuration resistance, allowing it to remain stable in air for a duration sufficient for device fabrication. After fabrication, the structures are immediately encapsulated with transferred 2D materials to isolate them from air, enabling long-term stability for photoelectric characterization. Our fabricated devices have been successfully stored in low-humidity air for over one month without performance degradation.

In short, the metal oxidation does not affect the performance of the device during our fabrication and measurement. For applications requiring extended shelf life or operation in harsher environments, encapsulation with an h-BN layer or other protective films can be readily implemented.

Revisions:

In the revised version, the comparisons between polycrystalline and single-crystalline Ag and relative discussions are added as the **Section 10** of **supplementary information**.

In the revised version, a paragraph “**For the stability of the single-crystalline Ag.....over one month without performance degradation.**” is added to the **Section 1** of **supplementary information**.

3. Considering the TAP dominates the photocurrent enhancement, and its efficiency scales with the fourth power of the input light's electric field, can the TAP process still be generated if the continuous wave (CW) light source is used for excitation, but not the pulsed laser used in present results? This point will affect the practicality and universality of this work.

Reply:

We appreciate the comprehensive and constructive comments.

As detailed in our response to **Comment 1** from Reviewer #1, the photocurrent of our device can be written as Eq. (1): $I_{\text{phc}} = k_1 \cdot P^{\alpha_1} + k_2 \cdot P^{\alpha_2}$, where $\alpha_1 \sim 1$ and $\alpha_2 \sim 2$. This model reflects the coexistence of two distinct photoresponse mechanisms: a linear process and a nonlinear process. At relatively low incident power, the photoresponse is dominated by the linear component ($\alpha_1 \sim 1$), which originates from the plasmonic resonance-induced hot-carrier injection. As the input power increases, the nonlinear TPA process ($\alpha_2 \sim 2$) becomes the dominant contributor to the photocurrent, as clearly evidenced by the power-dependent data in Fig. **R1**.

It is important to note that this nonlinear behavior is strongly dependent on the peak power density of the excitation source. Under CW illumination, the power density is substantially lower than that of a pulsed laser with the same average power. In such a scenario, the coefficient k_2 associated with the TPA process would be significantly suppressed, while k_1 , governing the linear hot-carrier response, would remain effective. Consequently, under CW operation, our device would functionally transition to a conventional hot-carrier photodetector relying on linear photon absorption.

In short, under CW illumination, the device responsivity decreases at higher input powers due to the dominance of the linear hot-carrier mechanism. And this point is supplied in the revised version. Nevertheless, near-infrared (NIR) photodetection remains functional across the entire operating power range.

Revisions:

In the revised version, the sentences “**Independent of the semiconductor's bandgap, hot carrier injection.....selectively discriminating optical signals based on chirality, phase, and polarization.**^{57,58}” are added to the ending of the “Performance of the photoelectric enhancement” part of **main text**.

4. The author compares the SHG signals of heterostructure on SiO₂/Si and on metasurface. Is it possible to generate SHG signal by the metasurface itself?

Reply:

We are grateful for highlighting this point, which is a crucial detail.

Although the metasurface itself generates SHG signal due to its inherent field enhancement, the signal is more than five orders of magnitude weaker than that produced by the TMDCs, as evidenced in Fig. R17. Given its negligible intensity relative to the dominant TMDC signal, the metasurface's direct SHG contribution was therefore omitted from the analysis in our manuscript.

Fig. R17 | **a,b**, SHG signal generated from the metasurface with (a) and without (b) MoS₂/WSe₂ heterostructure, 1550 nm input. Considering the differences of measuring conditions, the signal difference can be estimated as $\sim 1750 \text{ counts}/32 \text{ counts} \times 100 \text{ s}/1 \text{ s} \times (1 \text{ mW}/100 \mu\text{W})^2 \sim 540000$ folds.

Revisions:

In the revised version, a sentence “Note that, the metasurface itself can also generate SHG, which is significantly weak (more than five orders of magnitude weaker than that from TMDCs) and is therefore ignored in our analysis.” is added to the Line 12, Paragraph 2 of the “Performance of the photoelectric enhancement” part of main text.

5. The chiral-resolved photoresponse is achieved by further breaking the symmetry of the metasurface. Does it still support the anapole state after its symmetry is broken?

Reply:

We appreciate the valuable question. Our analysis confirms that it is selectively supported for the targeted left-handed circular polarization (LCP) but not for the right-handed (RCP).

The evidence, presented in Fig. 4g,h, shows a strong resonance and field enhancement only under LCP. The multipole decomposition in Fig. R18a under LCP illumination reveals the key signature of the anapole state: a destructive interference between ED and TD, alongside the hierarchy of $EQ > EO > ED+TD$. In contrast, the RCP response (Fig. R18b) does not fulfill this condition. The anapole state is absent, and the mode is primarily dominated by the ED, leading to a weak photoresponse.

Therefore, the symmetry breaking does not destroy the anapole state but rather makes it chiral-dependent, which is the very origin of the chiral-resolved photoresponse we report.

Fig. R18 | Multipole decomposition of the optical mode of chiral-resolved metasurface in Fig. 4 under LCP (a) and RCP (b) illumination.

Revisions:

In the revised version, the multipole decomposition of the chiral-resolved structures and the related discussion (including Fig. R18) are added as the Section 24 of supplementary information.

In the revised version, a sentence “For the multipole decomposition after breaking the symmetry along the x -direction, it does not destroy the anapole state but rather makes it chiral-dependent, as shown in Fig. S23.” is added to the ending of the Paragraph 1 in the “Design of chirality-resolved photodetector” part of main text.

References

1. Buscema, M. et al. Photocurrent generation with two-dimensional van der Waals semiconductors. *Chem. Soc. Rev.* **44**, 3691 (2015).
2. Xiong, Y.-F., Chen, J.-H., Lu, Y.-Q., & Xu, F. Broadband optical-fiber-compatible photodetector based on a graphene-MoS₂-WS₂ heterostructure with a synergetic photogenerating mechanism. *Adv. Electron. Mater.* **5**, 1800562 (2019).
3. Zhang, J. et al. Ultrafast saturable absorption of MoS₂ nanosheets under different pulse-width excitation conditions. *Opt. Lett.* **43**, 243–246 (2018).
4. Li, Y. et al. Giant two-photon absorption in monolayer MoS₂. *Laser & Photonics Rev.* **9**, 427–434 (2015).
5. Xu, C. & Webb, W. W. Measurement of two-photon excitation cross sections of molecular fluorophores with data from 690 to 1050 nm. *J. Opt. Soc. Am. B* **13**, 481–491 (1996).
6. Li, S. & She, C. Y. Two-photon absorption cross-section measurements in common laser dyes at 1.06 μm. *Optica Acta* **29**, 281–287 (1982).
7. Rumi, M. & Perry, J. W. Two-photon absorption: an overview of measurements and principles. *Adv. Opt. Photon.* **2**, 451–518 (2010).
8. Xie, Y. et al. Layer-modulated two-photon absorption in MoS₂: probing the shift of the excitonic dark state and band-edge. *Photonics Res.* **7**, 762–770 (2019).
9. Zhang, S. et al. Direct observation of degenerate two-photon absorption and its saturation in WS₂ and MoS₂ monolayer and few-layer films. *ACS Nano* **9**, 7142–7150 (2015).
10. Liu, C.-H., Chang, Y.-C., Norris, T. B. & Zhong, Z. Graphene photodetectors with ultra-broadband and high responsivity at room temperature. *Nat. Nanotech.* **9**, 273–278 (2014).
11. Lopez-Sanchez, O., Lembke, D., Kayci, M., Radenovic, A., & Kis, A. Ultrasensitive photodetectors based on monolayer MoS₂. *Nat. Nanotech.* **8**, 497–501 (2013).
12. Huo, N. & Konstantatos, G. Ultrasensitive all-2D MoS₂ phototransistors enabled by an out-of-plane MoS₂ PN homojunction. *Nat. Commun.* **8**, 572 (2017).
13. Koshelev, K., Favraud, G., Bogdanov, A., Kivshar, Y. & Fratallocchi, A. Nonradiating photonics with resonant dielectric nanostructures. *Nanophotonics* **8**, 725–745 (2019).
14. Baryshnikova, K. V., Smirnova, D. A., Luk'yanchuk, B. S. & Kivshar, Y. S. Optical anapoles: Concepts and applications. *Adv. Optical Mater.* **7**, 1801350 (2019).
15. Yezekyan, T., Zenin, V. A., Beermann, J. & Bozhevolnyi, S. I. Anapole states in gap-surface plasmon resonators. *Nano Lett.* **22**, 6098–6104 (2022).
16. Zenin, V. A. et al. Direct amplitude-phase near-field observation of higher-order anapole states. *Nano Lett.* **17**, 7152–7159 (2017).
17. Pardo, F., Bouchon, P., Haïdar, R. & Pelouard, J.-L. Light funneling mechanism explained by magnetoelectric interference. *Phys. Rev. Lett.* **107**, 093902 (2011).
18. Wang, Z. et al. Giant photoluminescence enhancement in tungsten-diselenide–gold plasmonic hybrid structures. *Nat. Commun.* **7**, 11283 (2016).
19. Zhang, Q.-H. et al. Electromagnetic Raman enhancement beyond gap limit. *Phys. Rev. Lett.* **134**, 136902 (2025).
20. Savinov, V., Fedotov, V. A. & Zheludev, N. I. Toroidal dipolar excitation and macroscopic electromagnetic properties of metamaterials. *Phys. Rev. B* **89**, 205112(2014).
21. Koshelev, K., Lepeshov, S., Liu, M., Bogdanov, A. & Kivshar, Y. Asymmetric metasurfaces with

- high-Q resonances governed by bound states in the continuum. *Phys. Rev. Lett.* **121**, 193903 (2018).
22. Hsu, C. W., Zhen, B., Stone, A. D., Joannopoulos, J. D. & Soljačić, M. Bound states in the continuum. *Nat. Rev. Mater.* **1**, 16048 (2016).
 23. Zhang, Q.-H. et al. Boosting optical nonlinearity of van der Waals materials with high-order multipoles. *Laser & Photonics Rev.* **19**, 2401850 (2025).
 24. X. Hu, T. Wang, L. Wang, and S. Dong, Surface-enhanced Raman scattering of 4-aminothiophenol self-assembled monolayers in sandwich structure with nanoparticle shape dependence: off-surface plasmon resonance condition. *J. Phys. Chem. C* **111**, 6962 (2007).
 25. Yu, Y. et al. Hot-carrier engineering for two-dimensional integrated infrared optoelectronics. *InfoMat.* **6**, e12556 (2024).
 26. Deng, J. et al. An on-chip full-Stokes polarimeter based on optoelectronic polarization eigenvectors. *Nat. Electron.* **7**, 1004–1014 (2024).
 27. Bu, Y. et al. Configurable circular-polarization-dependent optoelectronic silent state for ultrahigh light ellipticity discrimination. *Light Sci. Appl.* **12**, 176 (2023).
 28. Y. R. Shen, *The Principles of Nonlinear Optics*. Wiley-Interscience: New York, USA 1984.
 29. Zhou, F. & Ji, W. Two-photon absorption and subband photodetection in monolayer MoS₂. *Opt. Lett.* **42**, 3113–3116 (2017).
 30. Buscema, M. et al. Photocurrent generation with two-dimensional van der Waals semiconductors. *Chem. Soc. Rev.* **44**, 3691–3718 (2015).
 31. Sarma, R. et al. Broadband and efficient second-harmonic generation from a hybrid dielectric metasurface/semiconductor quantum-well structure. *ACS Photonics* **6**, 1458–1465 (2019).
 32. Huang, J. et al. Large-area synthesis of monolayer WSe₂ on a SiO₂/Si substrate and its device applications. *Nanoscale* **7**, 4193–4198 (2015).
 33. Dong, Z. et al. Broadband excitonic near-infrared photoresponse at the van der Waals heterostructure/metal interface. *ACS Photonics* **11**, 4209–4216 (2024).

To Reviewer #4:

Comments:

In this round of revision, the authors have addressed part of my concerns. However, several issues still remain and require further clarification or validation by the authors, for example, the first and third points.

Reply:

We sincerely thank the Reviewer for the continued engagement with our manuscript and for acknowledging that we have addressed part of the earlier concerns. We have carefully considered the remaining issues, and below we provide a point-by-point clarification and validation.

1. For the first comment, I suggest that the authors include a table offering a comprehensive and thorough comparison of key performance metrics among representative near-infrared photodetectors.

Reply:

We appreciate the constructive advice. As suggested, a comprehensive comparison of key performance metrics among representative near-infrared photodetectors is now provided in Table R1. The data clearly show that our MoS₂/WSe₂-based photodetector with an Ag metasurface achieves a high responsivity of 1.35 A/W at 1550 nm, surpassing most reported counterparts in similar wavelength ranges. Meanwhile, its broadband photoresponse further underlines the overall performance advantage of our device.

It is worth emphasizing the significant performance enhancement enabled by our plasmonic metasurface. By comparing the enhancement factor (EF) of our structure with those reported in other works, the metasurface supporting anapole states shows clear advantages, providing a substantial photocurrent enhancement of approximately 50000-fold, which further highlights the uniqueness and advancement of our structural design. Furthermore, the metasurface can be tailored to support chiral-resolved photodetection, which further extends the applicability and functionality of our proposed structure.

Table R1 | Photoelectric performances of structure-enhanced near-infrared photodetectors.

Material	Structural type	Wavelength (nm)	Delay (ms)	Responsivity (A/W)	EF	Reference
MoS ₂ /WSe ₂	Ag metasurface	1550	300	1.35	50000	This work
MoS ₂	Au nanoparticles	980	2.6	6.4×10^{-2}	14	1
MoS ₂	Ag nanoparticles	1550	–	5.39×10^{-4}	2.07*	2

MoS ₂	Au resonant wires	1070	216000	5.2	–	3
Si/ Gr	waveguide	1550	–	0.37	9.25*	4
BP	Plasmonic bowtie antenna	1550	–	1.42×10^{-2}	4	5
Colloidal QDs	Au metasurface	1530	2.32×10^{-3}	8.2×10^3	10	6
Si/ Gr	Au metasurface	1550	8×10^{-3}	1.5×10^{-5}	–	7

*Calculated from the data in the corresponding reference. Gr: Graphene; BP: black phosphorus; QD: Quantum Dot.

2. For the third comment, I believe the authors should further evaluate and characterize the device performance using a continuous-wave (CW) light source. If the responsivity cannot reach the level obtained under pulsed illumination, this limitation should be clearly clarified and discussed in the manuscript. This is because, for photodetector applications, illumination is predominantly continuous rather than pulsed in most practical scenarios.

Reply:

We are grateful for this valuable suggestion regarding the characterization of our device under CW illumination. In response, we have conducted measurements using a CW light source, and the results are presented in Fig. R1b.

As previously discussed, the photocurrent of our device follows the expression: $I_{\text{phc}} = k_1 \cdot P^{\alpha_1} + k_2 \cdot P^{\alpha_2}$, where $\alpha_1 \sim 1$ and $\alpha_2 \sim 2$. This model reflects the coexistence of two photoresponse mechanisms: a linear process ($\alpha_1 \sim 1$) attributed to plasmonic resonance-induced hot-carrier injection, and a nonlinear TPA process ($\alpha_2 \sim 2$). The power-dependent photoresponse under pulsed and CW illumination are shown in Fig. R1.

At low incident power ($< 100 \mu\text{W}$), the photoresponse under both CW and pulsed illumination is dominated by the linear mechanism, and the responsivities in the two regimes are comparable. As the incident power increases ($> 100 \mu\text{W}$), the response under pulsed illumination surpasses that under CW illumination. This is because the high peak intensity of the pulsed laser strongly promotes the nonlinear TPA process, whereas under CW illumination the response remains primarily linear due to the lower instantaneous power.

Consequently, the device responsivity under CW illumination remains stable at low power but decreases at higher power levels, as the nonlinear contribution is not fully activated. For instance, at an incident power of 2 mW, the responsivity under pulsed illumination reaches 1.35 A/W, while under CW illumination it is 0.12 A/W. Importantly, the device maintains functional near-infrared photodetection

across the entire measured power range.

Fig. R1 | Photocurrent of the device as a function of laser power at 1550 nm under (a) pulsed illumination and (b) CW illumination. The photocurrent can be fitted as $I_{\text{phc}} = k_1 \cdot P^{\alpha_1} + k_2 \cdot P^{\alpha_2}$. For pulsed illumination, $\alpha_1 = 0.86$, $\alpha_2 = 1.94$. For CW illumination, $\alpha_1 = 0.82$, $\alpha_2 = 0$.

References

1. Guo, J. et al. Near-infrared photodetector based on few-layer MoS₂ with sensitivity enhanced by localized surface plasmon resonance. *Appl. Surf. Sci.* **483**, 1037–1043 (2019).
2. Park, M. J., Park, K. & Ko, H. Near-infrared photodetector achieved by chemically-exfoliated multilayered MoS₂ flakes. *Appl. Surf. Sci.* **448**, 64–70 (2018).
3. Wang, W. et al. Hot electron-based near-infrared photodetection using bilayer MoS₂. *Nano Lett.* **15**, 7440–7444 (2015).
4. Goykhman, I. et al. On-chip integrated, silicon–graphene plasmonic Schottky photodetector with high responsivity and avalanche photogain. *Nano Lett.* **16**, 3005–3013 (2016).
5. Venuthurumilli, P. K., Ye, P. D. & Xu, X. Plasmonic resonance enhanced polarization sensitive photodetection by black phosphorus in near infrared. *ACS Nano* 2018, **12**, 4861.
6. Đorđević, N. et al. Metasurface colloidal quantum dot photodetectors. *ACS Photonics* **9**, 482–492 (2022).
7. Li, L. et al. Monolithic full-stokes near-infrared polarimetry with chiral plasmonic metasurface integrated graphene–silicon photodetector. *ACS Nano* **14**, 16634–16642 (2020).